# Utilizing the Drake Passage Time-series to understand variability and change in subpolar Southern Ocean pCO$_2$

Amanda R. Fay[1], Nicole S. Lovenduski[2], Galen A. McKinley[1], David R. Munro[2], Colm Sweeney[3-4], Alison R. Gray[5], Peter Landschützer[6], Britton B. Stephens[7], Taro Takahashi[1], Nancy Williams[8]

[1] Lamont Doherty Earth Observatory of Columbia University, New York, NY, USA
[2] Department of Atmospheric and Oceanic Sciences and Institute of Arctic and Alpine Research, University of Colorado, Boulder, CO, USA
[3] Cooperative Institutes for Research in Environmental Sciences, University of Colorado, Boulder, CO, USA
[4] NOAA Earth System Research Laboratory, Boulder, CO, USA
[5] School of Oceanography, University of Washington, Seattle, WA, USA
[6] Max Planck Institute for Meteorology, Hamburg, Germany
[7] National Center for Atmospheric Research (NCAR), Boulder, CO, USA
[8] College of Earth, Ocean, and Atmospheric Sciences, Oregon State University, Corvallis, OR, USA

**Abstract**

The Southern Ocean is highly under-sampled for the purpose of assessing total carbon uptake and its variability. Since this region dominates the mean global ocean sink for anthropogenic carbon, understanding temporal change is critical. Underway measurements of pCO$_2$ collected as part of the Drake Passage Time-series (DPT) program that began in 2002 inform our understanding of seasonally changing air-sea gradients in pCO$_2$, and by inference the carbon flux in this region. Here, we utilize available pCO$_2$ observations to evaluate how the seasonal cycle, interannual variability, and long-term trends in surface ocean pCO$_2$ in the Drake Passage region compare to that of the broader subpolar Southern Ocean. Our results indicate that the Drake Passage is representative of the broader region in both seasonality and long-term pCO$_2$ trends as evident through the agreement of timing and amplitude of seasonal cycles as well as trend magnitudes both seasonally and annually. The high temporal density of sampling by the DPT is critical to constraining estimates of the seasonal cycle of surface pCO$_2$ in this region, as winter data remain sparse in areas outside of the Drake Passage. An increase in winter data would aid in reduction of uncertainty levels. On average over the period 2002-2016, data show that carbon uptake has strengthened with annual surface ocean pCO$_2$ trends in the Drake Passage and the broader subpolar Southern Ocean less than the global atmospheric trend. Analysis of spatial correlation shows Drake Passage pCO$_2$ to be representative of pCO$_2$ and its variability up to several hundred kilometers away from the region. We also

compare DPT data from 2016 and 2017 to contemporaneous $pCO_2$ estimates from autonomous

biogeochemical floats deployed as part of the Southern Ocean Carbon and Climate Observations and

Modeling project (SOCCOM) so as to highlight the opportunity for evaluating data collected on

autonomous observational platforms. Though SOCCOM floats sparsely sample the Drake Passage region

for 2016-2017 compared to the Drake Passage Time-series, their $pCO_2$ estimates fall within the range of

underway observations given the uncertainty on the estimates. Going forward, continuation of the Drake

Passage Time-series will reduce uncertainties in Southern Ocean carbon uptake seasonality, variability, and

trends, and provide an invaluable independent dataset for post-deployment assessment of sensors on

autonomous floats. Together, these datasets will vastly increase our ability to monitor change in the ocean

carbon sink.

## 1. Introduction

The Southern Ocean plays a disproportionately large role in the global carbon cycle. Over the past few

decades, the ocean has absorbed approximately 26% of the carbon dioxide ($CO_2$) emissions from fossil fuel

burning and land use change [Le Quéré et al., 2016, 2017], and since the preindustrial era, the ocean has

been the primary sink for anthropogenic emissions [McKinley et al., 2017; Ciais et al., 2013]. The Southern

Ocean (south of 30°S) accounts for almost half of the total oceanic sink of anthropogenic $CO_2$ [Frölicher et

al., 2015; Gruber et al., 2009; Takahashi et al., 2009]. Though the importance of this region is widely

understood, the relative scarcity of surface ocean carbon-related observations in the Southern Ocean

hampers our ability to understand how this anthropogenic $CO_2$ uptake occurs against the background of

natural variability.

Observations and models suggest large variability in the strength of Southern Ocean $CO_2$ uptake on decadal

timescales. Several studies have reported a slow-down or reduction in the efficiency of Southern Ocean

$CO_2$ uptake from the 1980's to the early 2000's [Le Quéré et al., 2007; Lovenduski et al., 2008; Metzl

2009; Takahashi et al., 2012; Fay and McKinley, 2013; Lovenduski et al., 2015; Landschützer et al., 2014a,

2015a], followed by a substantial strengthening of the Southern Ocean $CO_2$ sink since 2002 [Fay and

McKinley, 2013; Fay et al., 2014; Landschützer et al., 2015a; Munro et al., 2015a; Xue et al., 2015].

Continued observational sampling efforts and coordination are required for quantifying and understanding

decadal changes in this important $CO_2$ sink region.

Initiated in 2002 and continuing to present, the Drake Passage Time-series is unique among Southern

Ocean research programs in both its spatial and temporal coverage. High-frequency underway observations

of the surface ocean partial pressure of $CO_2$ ($pCO_2$) are collected on the Antarctic Research and Supply

Vessel *Laurence M. Gould* on up to 20 crossings per year from the southern tip of South America to the

Antarctic Peninsula, spanning the Antarctic Circumpolar Current (ACC) and its associated Antarctic Polar

Front [Munro et al., 2015a, 2015b]. The DPT is also notable for sampling surface ocean $pCO_2$ during the

austral winter in all years from 2002 to present, providing valuable information about the full seasonal

cycle of $pCO_2$ in the poorly sampled Southern Ocean. Other ships have contributed observations in the

Drake Passage region including the *Polarstern* and the *Nathaniel B. Palmer*, however none have the

consistent temporal coverage as provided by the DPT.

The surface ocean $pCO_2$ observations from the DPT have provided the foundation for larger data sets,

which have been extensively used to examine variability and trends in $CO_2$ uptake in the broader Southern

Ocean [Fay and McKinley, 2013; Fay et al., 2014; Majkut et al., 2014; Landschützer et al., 2014b, 2015b;

Rödenbeck et al., 2015, Gregor et al., 2018]. In many of these studies, interpolated estimates of Southern

Ocean $pCO_2$ are used in conjunction with measurements of atmospheric $pCO_2$ to estimate variability and

trends in the air-sea $pCO_2$ gradient and, when combined with wind speed, air-sea $CO_2$ fluxes.


The physical oceanography of the Drake Passage region is unique in the Southern Ocean. Here, the strong

flow of the zonally unbounded ACC is funneled through a narrow constriction (~800 km), making it an

ideal location for sampling across the entire ACC system over a relatively short distance [Sprintall et al.,

2012]. At the same time, the unique nature of this circulation could potentially reduce the degree to which

the Drake Passage region is representative of the broader subpolar region. The DPT program takes

advantage of frequent *Gould* crossings to conduct physical and biogeochemical sampling of the ACC

system. Thus, before conclusions can be drawn about large-scale Southern Ocean carbon uptake and its

variability using data from the DPT, it is important to document how $pCO_2$ in this particular region compares with $pCO_2$ measured elsewhere in the subpolar Southern Ocean. In this study, we utilize

available ship-based surface ocean $pCO_2$ observations collected in the subpolar Southern Ocean to evaluate how the seasonal cycle, interannual variability, and long-term trends of surface ocean $pCO_2$ in the Drake Passage region compare to that of the broader subpolar Southern Ocean. Further, we highlight the opportunity for post deployment assessment of autonomous observational platforms passing through the Drake Passage utilizing the high frequency, underway $pCO_2$ measurements from the DPT.


## 2. Data

This study uses several observational datasets and data products of surface ocean $pCO_2$ in the Southern Ocean: measurements from the Surface Ocean $CO_2$ Atlas (SOCAT), which includes underway measurements from the DPT, interpolated estimates of the SOCAT data using a self-organizing map feed-

forward neural network (SOM-FFN) approach, and calculated $pCO_2$ estimates from biogeochemical Argo floats. While the SOCAT database reports the fugacity of carbon dioxide ($fCO_2$), for our analysis we consider datasets reporting $pCO_2$ and $fCO_2$ to be interchangeable. This is an acceptable assumption for surface ocean observations as $CO_2$ behaves closely to an ideal gas. Globally, the difference between these parameters is less than 2 μatm, with $fCO_2$ being smaller than $pCO_2$ by no more than 2 μatm due to

temperature dependence. This is roughly the reported uncertainty of shipboard observations of $pCO_2$ and well within the uncertainty of the observation-based $pCO_2$ estimates. Below, we describe each of these data sources in turn.

### 2.1 The Drake Passage Time-series (DPT)

A unique dataset of ongoing year-round observations beginning in 2002 is available from the Drake Passage Time-series. This data set provides an unprecedented opportunity to characterize the mean and time-varying state of the Drake Passage and surrounding waters using direct observations. In addition to high frequency underway observations of surface ocean $pCO_2$, other physical and biogeochemical variables measured onboard allow for a complete understanding of the carbonate system in the Drake Passage.

Analytical methods used to measure $pCO_2$ to ±2 µatm are described in detail by Munro et al. [2015a,

2015b].

**2.2 Surface Ocean CO$_2$ Atlas (SOCAT)**

SOCAT is a global surface ocean carbon dataset of $fCO_2$ values ($pCO_2$ corrected for the non-ideal behavior

of $CO_2$) [Sabine et al., 2013; Pfeil et al., 2013]. In this study, we utilize version 5 of this product

(SOCATv5) and include data with a reported WOCE flag of 2 and cruise flags A-D which results in a

dataset of roughly 18.5 million observations globally, spanning years 1957-2016, with uncertainties of ±2-5

µatm [Bakker et al., 2016]. This dataset includes over 740,000 observations contributed from the DPT.

Despite the large number of observations available in the Southern Ocean, data is spatially and temporally

concentrated, with strong seasonal biases. Most data are collected during reoccupations of supply routes to

Antarctic bases or on repeat hydrographic lines, which leaves large bands of the Southern Ocean

completely unsampled [Bakker et al., 2016].

**2.3 Self-Organizing Map Feed-forward Network Product (SOM-FFN)**

Landschützer et al., [2014b] use a two-step neural network approach to extrapolate the monthly gridded

SOCAT product in space and time. This results in reconstructed, basin-wide monthly maps of the sea

surface $pCO_2$ at a resolution of 1° × 1° [Landschützer et al., 2017]. Air–sea $CO_2$ flux maps are then

computed using a standard gas exchange parameterization and high-resolution wind speeds. The neural

network estimate is described and substantially validated in past publications [Landschützer et al., 2014,

2015a, 2016] and it was shown that the estimates fit observed $pCO_2$ data in the Southern Ocean with a root

mean square error (RMSE) of about 20 µatm and with almost no bias [Landschützer et al., 2015a,

supplementary material].

The SOM-FFN product used in this analysis was created from SOCATv5. Additionally, we generated an

alternative SOM-FFN product (SOM-FFN-noDP) using the same methodological setup but excluding the

$pCO_2$ data collected in the Drake Passage region for years 2002-2016, which represents the years of the

DPT program.

**2.4 SOCCOM Floats**

The Southern Ocean Carbon and Climate Observations and Modeling (SOCCOM) project

(http://soccom.princeton.edu) aims to deploy approximately 200 biogeochemical profiling floats over a

five-year period (2015 to 2020) in an effort to fill observational gaps in the Southern Ocean. In total, over

100 floats carrying some combination of additional biogeochemical sensors (i.e., pH, nitrate, oxygen,

fluorescence, and backscattering) have been collecting data since April 2014 [Johnson et al., 2017]. With

the float's capability to measure pH and utilization of existing algorithms for predicting total alkalinity,

$pCO_2$ can be calculated from the collected observations and compared to underway observations [Williams

et al., 2017].

The uncertainty range for these calculated $pCO_2$ values is estimated to be 2.7% (±11 µatm at 400 µatm) and

takes into account multiple sources of uncertainty including measurement error, uncertainties introduced

through the quality control procedures, and uncertainties in seawater carbonate system thermodynamics

[Williams et al., 2017]. $pCO_2$ estimates from profiling floats have not been included in the SOCAT

database because they do not directly measure surface water $CO_2$. For consistency, we maintain this

separation in our analysis and limit our study of SOCCOM floats to direct comparisons to DPT values in

Section 5.

**3. Methods**

The SOCATv5 database from 2002-2016 is considered here to match the years of overlap with DPT

observations, which began in 2002. The SOCAT dataset is then subsampled to include only observations

with reported salinity values in the 33.5 - 34.5 range and a distance-to-land value greater or equal to 50 km.

This step restricts our analysis to open-ocean observations, since coastal observations report lower salinity

values, which correspond to low $pCO_2$ values due to the influence of fresh water and ice melt. SOCCOM

float files were downloaded on 02 April 2018 and reported $pCO_2$ values, are an average of all data collected

in the top 20m of water, calculated using alkalinity derived from the LIAR algorithm [Carter et al., 2016],

to remain consistent with previous SOCCOM float analysis.

The Southern Ocean region of interest is the Southern Ocean Subpolar Seasonally Stratified (SPSS) biome as defined in Fay and McKinley [2014] as the region of the Southern Hemisphere with climatological SST <8°C but excluding areas with a sea-ice fraction greater than or equal to 50% (Figure 1). While the SPSS

biome encompasses the Drake Passage, we further define a Drake Passage region as the portion of the Southern Ocean SPSS biome bounded by 55°W and 70°W lines of longitude (Figure 1, black box). This is similar to the region analyzed in Munro et al [2015a] however it extends the region of interest to the northern and southern extents of the SPSS biome.

In order to compare the seasonal cycle and long-term trends in the Drake Passage with the broader SPSS biome, we analyze surface ocean $pCO_2$ from 3 subsets of the SOCAT database: SOCAT-all which includes all available SOCATv5 data from 2002-2016 in the SPSS biome, SOCAT-DP which includes SOCATv5 data within the longitudinally-defined Drake Passage region (Figure 1, with 62% of this data obtained by the LDEO/Univ. Colorado group), and SOCAT-noDP which excludes any data within the longitudinally-

defined Drake Passage region of the SPSS biome. All datasets are first averaged to monthly, 1°x1° resolution. Monthly means are then calculated for the SPSS biome by first removing the background mean annual climatological value of $pCO_2$ at each 1° x 1° location [Landschützer et al., 2014a] to aid in accounting for the potential of spatial aliasing in the sparsely sampled Southern Ocean [Fay and McKinley, 2013].


Alternate definitions of the larger Southern Ocean region of interest were considered during our analysis, including a subdivision of the SPSS into a Northern SPSS and Southern SPSS, with the boundary defined by the location of the mean position in the Antarctic Polar Front [Freeman and Lovenduski 2016; Freeman et al., 2016; Munro et al., 2015b]. As discussed in Munro et al., [2015a], Drake Passage Time-series

observations north of the front report higher $pCO_2$ values than to the south and they find a larger trend in $pCO_2$ in the north for years 2002-2015. Additionally, the seasonal cycle amplitude north of the front is much larger and well defined than south of the front. We see these patterns in the SOCAT dataset as well; however, given the goal of this research, we choose to consider the entire north-to-south extent of the SPSS

as a whole. Outside of the Drake Passage region, available data is limited such that analysis over north and

south subregions would be impossible.

Additionally, we consider analysis over the Polar Antarctic Zone (PAZ), defined as the area between the

Subantarctic Front and the sea ice zone [Williams et al., 2017] (Supplementary Figure 1). While differences

exist in trends and seasonality when using the PAZ definition (Supplementary Figures 2-3), the overall

conclusions of the relationship between SOCAT-DP and SOCAT-all remain largely unchanged when using

this alternate regional definition.

Biome-scale monthly means are compared and used to calculate seasonal cycles and trends. Seasonal

cycles are calculated by first removing a 1.95 $\mu$atm yr$^{-1}$ trend to account for increasing atmospheric $CO_2$

during the 2002-2015 period [Dlugokencky et al., 2015]. Seasonal uncertainties (Figure 2) are estimated as

1 standard error from the mean of all available biome mean values for a given month. This is a conservative

estimate of the uncertainty in any given month because of inconsistent annual coverage and spatial

undersampling biases. Reported trends are calculated by fitting a single harmonic and linear trend to the

biome-scale monthly means as done in Fay and McKinley [2013]. Trends are not statistically different if

the calculated mean seasonal cycle is removed instead of the choice to fit a harmonic to the data. Seasonal

trends are calculated with a simple linear fit to the seasonal monthly means.

**4. Results and Discussion**

**4.1 Seasonal cycle**

The mean seasonal cycle of $pCO_2$ (corrected to reference year 2002) in the Southern Ocean SPSS biome for

the 3 SOCAT datasets and the full SOM-FFN estimate indicate broad agreement (Figure 2). Here, surface

ocean $pCO_2$ levels reach a maximum in austral winter (June to August), when deep mixing delivers carbon-

rich water to the surface, and a minimum in austral summer (December to February), when biological

production draws down the inorganic carbon from the surface [Takahashi et al., 2009]. Temperature also

plays a role in modulating the $pCO_2$ seasonal cycle in the Southern Ocean. Winter cooling drives $pCO_2$

lower at the same time as deep winter mixing elevates surface carbon levels. During the summer, warming

temperatures raise $pCO_2$ while biological utilization of carbon drives surface $pCO_2$ levels lower [Munro et al., 2015b].

The average amplitude of the detrended seasonal cycle of $pCO_2$ (max-min) is 23 μatm (Figure 2), smaller than the high latitude oceans in the Northern Hemisphere [Takahashi et al., 2002, 2009; Landschützer et al., 2015b]. The small amplitude of the $pCO_2$ seasonal cycle in this region is due to the similar magnitude and opposite phasing of temperature and carbon supply/utilization effects [Munro et al., 2015b]. In all months, mean surface ocean $pCO_2$ levels in the Southern Ocean SPSS are below atmospheric which ranges from a

global annual mean of 372 ppmv in 2002 to 399 ppmv in 2015, indicating that this region has been a persistent $CO_2$ sink over the period of analysis [Dlugokencky & Tans, 2017].

Figure 2 also shows the uncertainty of the seasonal mean, with shading representing 1 standard error from the monthly mean for each dataset, defined as the standard deviation divided by the square root of the

sample size (here, number of years with available data in that month). Uncertainty estimates vary for each month of the seasonal cycle with a minimum uncertainty of 1.1 μatm (June, SOCAT-DP) to a maximum of 5 μatm (July, SOCAT-DP). These estimates are of the same magnitude as the measurement accuracy of underway $pCO_2$ in SOCAT (±2-5 μatm).

Figure 3 indicates how diverse Southern Ocean $pCO_2$ data density is in space and time. Compared to the regular sampling of the DPT, there are many fewer repeated occupations of SR03 south of Australia [Shadwick et al., 2015], along the Prime Meridian [Hoppema et al., 2009; Van Heuven et al., 2011], and in the southwestern Indian sector [Metzl et al., 1999; Lo Monaco et al., 2005, 2010; Metzl, 2009]. Specifically, during austral winter, data availability outside of the Drake Passage region is extremely

limited due to the few ships operating in winter and the difficult conditions that the wintertime Southern Ocean presents to data collection efforts (Figure 3b).

Despite irregular sampling, average seasonal cycles of the 3 SOCAT datasets are quite similar, with few statistically significant differences given the uncertainty bounds. SOCAT data from the Drake Passage

region (SOCAT-DP, gray) exhibits relatively large estimated uncertainty (average for all months = 2.22

μatm), despite the frequent coverage and smaller region considered. This indicates that large interannual

variability is inherent to the Drake Passage region, especially in the well-observed austral summer months.

Despite data being much more regularly collected in this region than in the rest of the Southern Ocean

(Figure 3), there are still months of quite limited observations, specifically July and August (Figure 2).

SOCAT-all has monthly uncertainties averaging 1.7 μatm with the largest uncertainties in January and July

(Figure 2, blue). Data availability for SOCAT-all is consistent for much of the year, with most months

having observations in at least 13 of the 15 years considered in this analysis (Figure 2). The exceptions are

July and August that have data from only 8 and 10 years, respectively.

The SOCAT-noDP seasonal cycle is similar to that of the other datasets but deviates in the austral

fall/winter, specifically May and June. In winter, SOCAT-noDP suggests higher $pCO_2$ than SOCAT-DP or

SOCAT-all, though the limited data in June and July must be considered when drawing conclusions from

this difference (Figure 2, 3b). With June and July data available for fewer than 5 of the 15 years covered in

the analysis it is possible that the peak shown here could be biased by the few years included, specifically

for the month of June. In contrast, SOCAT-DP has data for nearly all of the years considered in these

months. The data that is available during May and June in SOCAT-noDP is from regions downstream of

the Drake Passage (Figure 3b).

Seasonal cycles are consistent when analyzing the PAZ region (Supplementary Figure 2), however the

SOCAT-DP seasonal cycle exhibits two maxima possibly due to the omission of the southern area of the

Drake Passage (Supplementary Figure 1) which would cause values for the PAZ region to be greater than

those shown for the DP region of the SPSS. The June peak in SOCAT-noDP also remains when

considering the PAZ region. Amplitudes are comparable given the uncertainty, however the seasonal

amplitude for each dataset is slightly larger over the SPSS biome than the PAZ, likely due to the more

northern expansion of the PAZ region downstream of the Drake Passage and the exclusion of the southern

Drake Passage region in the boundary of the PAZ.

Overall, given available data, the seasonal cycles are statistically indistinguishable for data collected inside

and outside of the Drake Passage region, for all months with at least 5 years of observations (Figure 2).

This analysis of SOCAT $pCO_2$ data indicates that the Drake Passage seasonal cycle is representative of the

broader SPSS biome seasonality, based on the available observations to date, but increased observations

outside of the Drake Passage during May and June are needed to provide a more robust comparison.

Additionally, the seasonal cycles from all 3 SOCAT datasets closely resemble the smoothed seasonality of

the interpolated SOM-FFN product in the SPSS biome (Figure 2). Sparse sampling outside of the Drake

Passage during winter months leads to this estimated seasonal cycle of SOCAT-all being driven by Drake

Passage data. Enhanced wintertime data collection, especially in regions outside of the Drake Passage, is

required to better constrain the full seasonal cycle of surface ocean $pCO_2$ in the Southern Ocean SPSS.

### 4.2 Interannual variability

The high resolution of the time-series data in the Drake Passage allows for close examination of temporal

variability in $pCO_2$ with relatively low uncertainty [Munro et al., 2015a]. We investigate the interannual

variability in Drake Passage $pCO_2$ in Fig. 4a, where deseasonalized and detrended anomalies [Fay and

McKinley, 2013] from the SOCAT-DP dataset are shown in gray, with the black line representing these

anomalies smoothed with a 12-month running mean. Over the 2002-2016 period, the variance in $pCO_2$

anomalies is 66 $\mu atm^2$. Monthly anomalies are as large as ±30 $\mu atm$, and 12-month smoothed anomalies as

large as ±12 $\mu atm$ in this dataset.

A model-based study by Lovenduski et al., [2015] finds interannual variability in $pCO_2$ to be low in the

Drake Passage compared to other Southern Ocean regions for years 1981-2007. In contrast, we find that

detrended and deseasonalized anomalies from SOCAT-noDP and SOCAT-DP have comparable variances

(59 $\mu atm^2$ and 66 $\mu atm^2$). This result, however, is likely strongly affected by the previously discussed

seasonal data gaps outside of the DP region or potentially by the different years considered in these two

analyses. Conducting a similar analysis of the reported SOCAT sea surface temperature (SST) values does

find the variance for SOCAT-DP to be significantly lower than SOCAT-noDP (0.93$°C^2$ and 2.72$°C^2$

respectively). As the same sampling issues exist for SST as for $pCO_2$ in SOCAT, an alternate method to

address this issue is needed to resolve these conflicting results.

The SOM-FFN data product offers complete seasonal and regional coverage, and thus the comparison of variance in Drake Passage to all the Southern Ocean can be made in this context. Results for SOM-FFN are

different from both the SOCAT findings above and the results of Lovenduski et al., [2015]. For the SPSS biome area of SOM-FFN $pCO_2$, the variance of detrended and deseasonalized anomalies is significantly higher within the Drake Passage region than outside of the region (14.2 $\mu atm^2$ and 6.2 $\mu atm^2$, respectively). It should be noted that variances are significantly lower for the SOM-FFN because of its interpolation. We are left without a clear picture as to whether Drake Passage is more or less variable in $pCO_2$ than the rest of

the Southern Ocean SPSS. This conundrum is clearly due to the lack of data availability, particularly outside the Drake Passage during winter months (Figure 3b).

Given the lack of data, the degree to which the Drake Passage represents interannual variability within the Southern Ocean SPSS can only be considered in the context of the SOM-FFN data product. To produce

independent estimates of correlations between Drake Passage and other points, we use a version of the SOM-FFN product created without the inclusion of any observations in our defined Drake Passage region (SOM-FFN-noDP, Figure 4b), and assess correlations to SOCAT data within the Drake Passage. Anomalies have been detrended and deseasonalized, and grayed areas indicate that the correlation is not significant at the 95% confidence level (Figure 4b). The strongest positive correlations are within Drake

Passage, upstream of the Drake Passage into the central Pacific SPSS, and in the Indian Ocean sector of the SPSS biome (Figure 4b). Weaker positive correlations are found in the western Pacific SPSS, as well as a few areas in the Atlantic sector of the SPSS. No regions of widespread strong negative correlations are observed in the SPSS biome. This is consistent with the analysis of Munro et al., [2015b] who estimate the footprint of the Drake Passage extending upstream into the eastern Pacific sector of the ACC.


**4.3 Trends, 2002-2016**

Trends for all data (annual), as well as summer (DJF) and winter (JJA), are estimated from the three SOCAT datasets, the SOM-FFN data product, and the SOM-FFN product subsampled as SOCAT-DP

(SOM-FFN-sampled), in all cases following the approach of Fay & McKinley [2013]. Similar to the

climatological $pCO_2$ seasonal cycle, annual trends for the 3 SOCAT datasets are indistinguishable given the

68% confidence intervals (Figure 5, Supplementary Table 1).

All annual trends are less than the 2002-2016 atmospheric $pCO_2$ trend of 1.95 µatm yr$^{-1}$ [Dlugokencky et

al., 2015], indicating that the Southern Ocean has been a growing sink for atmospheric carbon over 2002-

2016 (Figure 5, far left). Comparing the different estimates, SOCAT-DP (gray bar) and SOCAT-all (blue

bar) have annual trends slightly below that of the full SOM-FFN, however with greater uncertainty bounds.

The annual trend from the SOCAT-all dataset (blue) is nearly identical to the SOCAT-DP trend in both

mean and uncertainty. These are not statistically different from the SOCAT-noDP, although the SOCAT-

noDP dataset does yield a slightly larger annual trend. While SOCAT-noDP yields the largest annual trend

of the 3 datasets, it still falls well below the atmospheric trend. These trends are comparable to those

reported in Munro et al., [2015b], Takahashi et al., [2012], and Fay et al., [2014] despite these studies

utilizing different datasets, methods, and regional boundaries. Takahashi et al., [2014], similar to Munro et

al., [2015b], show that trends in the northern portion of the Drake Passage are greater than those south of

the front. Our analysis of regions north and south of the front confirms this (not shown).


Sampling the SOM-FFN data product as the SOCAT-DP dataset (SOM-FFN-sampled) is one way to

estimate the impact of the available data coverage in the Drake Passage region as compared to the

hypothetical situation of perfect data coverage in the SPSS biome. Sampling lowers the trend, making it

significantly smaller than the full SOM-FFN trend. This reduction leads to an annual trend very similar to

that of SOCAT-DP and SOCAT-all. This conclusion emphasizes the need for increased observations

around the Southern Ocean as it implies we are potentially not accurately capturing the true trend in this

region with the available data coverage.

Conclusions of these comparisons are largely maintained for summer and winter trends (Figure 5, center

and right). Uncertainty increases when considering seasonal trends due to reduced data quantity. All trends

are statistically indistinguishable for summer months; however, the SOCAT-DP trend shows the largest

change from the reported annual trends. For winter, SOCAT-noDP is not shown because unlike SOCAT-all and SOCAT-DP, not all years have available data during this season (Figure 2). Overall, winter trends are slightly higher than summer trends. Even given the uncertainties, winter and summer trends are clearly

distinguishable for SOCAT-DP, SOCAT-all, and the full and sampled SOM-FFN product. In each of these datasets, the winter trend is roughly 0.5 µatm yr$^{-1}$ higher than the summer trend. While winter trends have larger differences and larger uncertainties, consistent with reduced data availability, this seasonal difference in trends is significant. Further and more detailed consideration of this seasonal comparison is warranted. Initial investigations indicate that 2016 had anomalously high wintertime $pCO_2$ values (not shown).

Specifically when trends are calculated with the same datasets for 2002-2015, winter trends are significantly lower than the atmospheric trend (SOCAT-DP: 1.53±0.32 µatm yr$^{-1}$, SOCAT-all: 1.59±0.27 µatm yr$^{-1}$, SOM-FFN: 1.70±0.09 µatm yr$^{-1}$). It is important to consider the impact of anomalous values at the end of selected time-series, specifically for time-series less than 15 years [Fay & McKinley, 2013].

An investigation of trends from the full SOM-FFN product and that of the SOM-FFN-noDP product for the entire Southern Ocean SPSS biome for years 2002-2016 indicates an increasing carbon uptake by the ocean with some interannual variability (Figure 6). If the Drake Passage data is omitted during the creation of the product (SOM-FFN-noDP), carbon flux and $pCO_2$ trends are unchanged (Figure 6). Both estimates illustrate that for 2002-2016, the Southern Ocean SPSS biome was an important sink of carbon dioxide.

Trend analysis for the PAZ region (Supplementary Figure 3, Supplementary Table 1) produces comparable results. Annual trends are indistinguishable between the 3 SOCAT datasets as well as between the SOM-FFN product, both full and sampled. It could be that the greater extent of the PAZ northward yields better agreement between the datasets and the SOM-FFN product. All annual trends are also below the

atmospheric trend. Summer and winter trends for the PAZ are consistent with results for the SPSS biome with winter trends being larger than summer trends, most significantly for the SOCAT-DP and SOM-FFN datasets. While actual trend values are different from those shown in Fig. 2, the results show that the relationship between trends for the 3 SOCAT datasets are indistinguishable for both seasons and annual analyses.

**5. DPT as a pCO$_2$ evaluation point for biogeochemical profiling floats**

Starting in late 2014, autonomous biogeochemical profiling floats have been deployed as part of the

SOCCOM project, and as of December 2017, ten floats had traveled through or were approaching the

Drake Passage region (Figure 7a). These floats offer a new opportunity to complement our oceanographic

understanding that has been developed primarily with traditional shipboard observations. Results above

show that a lack of observations outside of the Drake Passage region may contribute to the large

uncertainties in both seasonality and trends, which limits the conclusions we are able to make with

currently available shipboard data. As floats provide autonomous, near real-time observations covering

existing spatial and temporal gaps throughout the Southern Ocean and ship-based systems provide high

density observations at higher accuracy (±2.7% or 11 µatm at a pCO$_2$ of 400 µatm for floats compared to

±2 µatm for ships), there is great potential for these two platforms to work in concert to provide a whole

Southern Ocean carbon observing system. However, there are limitations of float observations notably the

indirect estimate of pCO$_2$ from pH and the requirement to adjust the sensor calibrations post-deployment by

reference to deep (near 1500 m) pH values estimated from multiple linear regression equations fitted to

high quality, spectrophotometric pH observations made on repeat hydrography cruises [Williams et al.,

2016; 2017; Johnson et al., 2016; 2017]. Further comparisons between float-estimated pCO$_2$ and shipboard

observations is clearly warranted, and the complementary strengths of the Drake Passage Time-series make

it an ideal dataset to help address these issues.

Here we utilize the underway Drake Passage Time-series pCO$_2$ data to conduct comparisons to nearby

SOCCOM floats, considering both seasonality as well as fine-scale crossovers. Note that in this section of

the analysis we utilize data only from the Drake Passage Time-series (available at

nodc.noaa.gov/ocads/data) instead of the SOCATv5 dataset because SOCAT data are not available after

2016 at the time of writing.


A strong benefit of autonomous observation systems is their ability to sample regions and times that are not

often surveyed by ships. SOCCOM floats collect data throughout the year, and especially important are the

additional observations in austral winter, a time when there are limited opportunities for ship-based

measurements. While currently only two full years of data are available from floats within the Drake

Passage (2016-2017) they span the full width of the region (Figure 7a) and are able to observe during each

month of the year. A seasonal comparison of monthly mean $pCO_2$ values for the DPT data and float $pCO_2$

estimates within the defined Drake Passage region show that both platforms capture the expected seasonal

cycle for the subpolar Southern Ocean with a wintertime peak and summertime low (Figure 7b). All

datasets shown have been adjusted to 2017 using the mean atmospheric trend (1.95 μatm yr$^{-1}$)

[Dlugokencky et al., 2015], and thus mean values are higher than shown in the seasonal curves of Fig. 2.

Standard error shading on the seasonal cycles (Figure 7b) includes considerations of measurement accuracy

as this differs substantially between these two platforms. Shading represents 1 standard error accounting for

the spatial and temporal heterogeneity of the sample and measurement error (2.7% or ±11 μatm at a $pCO_2$

of 400 μatm for floats; ±2 μatm for DPT data), combined using the square root of the sum of squares.


The seasonal cycle derived from float-estimated $pCO_2$ has a larger seasonal amplitude compared to the

DPT data from 2002-2017, due to an earlier and much lower observed summertime minimum. The

difference in summertime minima is smaller however when DPT data from only 2016 and 2017 are

considered (Figure 7b). The remaining difference between the floats and the DPT 2016-2017 data might be

an artifact of the specific locations sampled, as floats and ships are not exactly synchronous, as well as the

conditions specific to 2016 and 2017. In summer 2016 for example, the floats appear to have captured a

strong phytoplankton bloom to the north and upstream of the Drake Passage, not captured by the DPT, that

resulted in strong inorganic carbon uptake and low $pCO_2$; the floats did not sample in the southern region

where $pCO_2$ is substantially higher in the early spring (Figure 8). However, there is no indication from the

2002-2017 seasonal cycle that this low excursion of the $pCO_2$ persists when looking at the entire Drake

Passage region (Figure 7b). Since phytoplankton blooms typically progress southward during spring

[Carranza and Gille, 2015] this difference in phasing likely results from the floats sampling preferentially

the earlier northern uptake.

Underway Drake Passage Time-series $pCO_2$ data in the SPSS biome has a large range, often spanning over

100 µatm each month as shown in the time-series in Fig. 8, largely related to the 10°C temperature gradient and associated physical and biological dynamics that are captured over the region. $pCO_2$ from all floats east of 90°W and west of 55°W are also plotted on the time-series (Figure 8, diamonds) with the reported $pCO_2$ value being an average for all depths shallower than 20 m. Float-based $pCO_2$ surface estimates largely fall

within the range of the direct underway $pCO_2$ observations, however notable differences do exist when spatial and temporal differences are taken into consideration. Float estimates from the central Drake Passage in winter (JJA) 2017 (Figure 8b) are higher than nearby DPT observations, though cruise data do not precisely overlap in time. Overall, the range of the DPT observations are far larger than the range of estimated $pCO_2$ from floats inside the Drake Passage region because they regularly span across the full

width of the Drake Passage where meridional decorrelation length scales are relatively short [Eveleth et al., 2017]. Conversely, floats tend to sample along the path of the ACC.

As floats offer autonomous, frequent observations and ships offer data of the highest quality, it is ideal for these two platforms to work in partnership. Analysis of direct comparisons between DPT data and

SOCCOM floats at crossover points indicates more precisely this potential (Figure 9). As of December 2017, there have been six occurrences of floats surfacing near DPT observations within a window of 75 km, 3 days, and have a reported SST within 0.3°C of each other (Figure 9a). This window is consistent with the crossover criteria used by the SOCAT community to quality control shipboard data [Pfeil et al., 2013; Olsen et al., 2013]. Figure 9a shows locations of the floats and the nearby DPT observations that fit this

crossover window. As DPT offers high frequency observations, all available measurements over the 3-day window are shown (Figure 9b). Also indicated are DPT observations that crossover within a 50km and 2-day window and 25km and 1-day window (Figure 9b, black 'x' and squares respectively), both also with the 0.3°C SST criteria.

This comparison of the calculated $pCO_2$ from the floats and observed DPT $pCO_2$ reveals a broad correspondence (passing through the 1:1 line) in all 6 crossover instances within the ±2.7% relative standard uncertainty of the SOCCOM float measurements and ±2 µatm DPT uncertainty (Figure 9b shading). While all float crossovers do intersect the 1:1 line given their stated uncertainties, these

comparisons reveal the large range of $pCO_2$ captured by high-frequency shipboard measurements in a

relatively small region and illustrates that this range cannot be fully captured by floats surfacing only once

every 10 days. Further investigation of crossovers in the entire Southern Ocean region is needed; the DPT

provides the most likely occurrence for this, although other regions with frequent ship traffic and

autonomous platforms with biogeochemical capabilities should also be utilized when feasible. Additional

post-deployment data quality checks using the underway surface $pCO_2$ data from DPT and other ship-based

programs should be conducted, and more thorough assessments could be achieved if hydrocast

observations were planned to occur in the vicinity of a passing biogeochemical float. Such coordinated

efforts would significantly advance monitoring of the carbon cycle in the Southern Ocean.

**6. Conclusions**

The Drake Passage Time-series illustrates the large variability of surface ocean $pCO_2$ and exemplifies the

value of sustained observations for understanding changing ocean carbon uptake in the Southern Ocean.

This is the only location where carbon measurements throughout the entire annual cycle in the subpolar

Southern Ocean have been made regularly over the past two decades. The available observations to date

indicate that the Drake Passage seasonal cycle is representative of the seasonality observed for the entire

SPSS biome, but increased observations outside of the Drake Passage, specifically during austral winter,

are needed to provide a more robust comparison. Uncertainties in the seasonality for all datasets studied

remain considerable given the dynamic nature of this region and the short time-series considered.

Specifically, a lack of winter data in all years limits the direct conclusions for differences between the

Drake Passage and the larger SPSS biome where we see a discrepancy in the timing of the winter maxima.

These findings can direct specific goals for focus regions of future observations. Specifically, insufficient

wintertime data in regions outside of the Drake Passage limits our assessment of how representative Drake

Passage data is of the larger subpolar region.

The magnitude of interannual variability is comparable for SOCAT $pCO_2$ data within and outside of the

Drake Passage region of the SPSS biome, a finding that conflicts with results from previous modeling and

analysis of the SOM-FFN product. A clear idea of whether the Drake Passage is more or less variable in

pCO$_2$ will require increased data, particularly during the austral winter, outside of the Drake Passage.

Given these data restrictions, the representativeness of the larger SPSS biome is also investigated using the

SOM-FFN product. Within this gap-filled data product, monthly anomalies in the Drake Passage region are

representative of broad swaths of the Southern Ocean, specifically regions upstream of the Drake Passage,

but strong relationships are also evident in regions in the Indian Ocean sector of the Southern Ocean.

Consistent with this finding, estimates of long-term trends do not change substantially if observations in the

Drake Passage are removed from the SOM-FFN analysis. Across approaches to data analysis, trends in

annual oceanic pCO$_2$ trends for 2002-2016 are less than the atmospheric pCO$_2$ trend, confirming previous

findings that the Southern Ocean has, on average, been a growing sink for atmospheric carbon over this

period.

Comparisons between underway DPT measurements and SOCCOM float estimates taken within the Drake

Passage show broad agreement, while a fine-scale crossover investigation demonstrates their direct

correspondence given uncertainty ranges for SOCCOM float pCO$_2$ estimates. Continuation of high-

temporal measurements of the DPT in addition to expanded programs to target floats with both underway

observations and frequent hydrocasts serving as independent datasets for post-deployment, will provide

high-value comparisons, improving community confidence in float-based pCO$_2$ estimates. Coordinated

monitoring efforts that combine a well-calibrated array of autonomous biogeochemical floats with a robust

ship-based observational network will improve and expand monitoring of the carbon cycle in the Southern

Ocean in the future.

**Acknowledgements.** We are grateful for funding from NSF (PLR-1543457, OCE-1558225, OCE-1155240), NOAA (NA12OAR4310058), and NASA (NNX17AK19G). NCAR is sponsored by the National Science Foundation. We acknowledge support from the Space Science and Engineering Center of University of Wisconsin – Madison and Columbia University. The authors are especially grateful for the efforts of the marine and science support teams of the ARSV Laurence M. Gould, particularly Timothy Newberger, Kevin Pedigo, Bruce Felix, and Andy Nunn. Underway DPT measurements presented in this manuscript are archived at NOAA's National Centers for Environmental Information (https://www.nodc.noaa.gov/ocads/oceans/VOS_Program/LM_gould.html). The Surface Ocean CO$_2$ Atlas (SOCAT) is an international effort, supported by the International Ocean Carbon Coordination Project (IOCCP), the Surface Ocean Lower Atmosphere Study (SOLAS), and the Integrated Marine Biogeochemistry and Ecosystem Research program (IMBER), to deliver a uniformly quality-controlled surface ocean CO$_2$ database. The many researchers and funding agencies responsible for the collection of data and quality control are thanked for their contributions to SOCAT. Float data were collected and made freely available by the Southern Ocean Carbon and Climate Observations and Modeling (SOCCOM)

Project funded by the National Science Foundation, Division of Polar Programs (NSF PLR-1425989), supplemented by NASA, and by the International Argo Program and the NOAA programs that contribute to it. (http://www.argo.ucsd.edu, http://argo.jcommops.org). The Argo Program is part of the Global Ocean
Observing System.

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

Figure Captions

Figure 1

Map of Subpolar seasonally stratified (SPSS) biome [Fay and McKinley 2014], defined at 1° x 1°

resolution. The red line represents the mean location of the Antarctic Polar Front [Freeman and

Lovenduski, 2016], interpolated to a 1° x 1° grid. The black box represents the Drake Passage region

considered in this analysis.

Figure 2

Mean surface ocean $pCO_2$ seasonal cycle estimate for years 2002-2016, for the SPSS biome from each

dataset, shown on an 18-month cycle, calculated from a time-series corrected to year 2002 (atmospheric

trend of 1.95 μatm yr$^{-1}$ removed). Shading represents 1 standard error for biome-scale monthly means

driven by interannual variability; there is no error represented for SOM-FFN. Bar plot indicates the number

of years containing observations in a given month (maximum of 15 years) for the SOCAT-DP SOCAT-

noDP, and SOCAT-all datasets.

Figure 3

Data density of $pCO_2$ observations from the SOCATv5 dataset within each 1° x 1° gridcell. Data is

restricted to years 2002-2016. Salinity values outside of 33.5-34.5 psu and observations within 50 km of

land are omitted. (a) data from all months of the year; (b) data from only June, July, and August (austral

winter). Gray lines designate boundary of SPSS biome and Drake Passage region for reference.

Figure 4

(a) Temporal evolution of deseasonalized, detrended monthly SOCAT-DP $pCO_2$ anomalies (gray bars) over

2002-2016, with 12-month running averages (black line) overlain. (b) Correlation between monthly

SOCAT-DP $pCO_2$ anomalies and the $pCO_2$ anomalies estimated from the SOM-FFN-noDP product

(created without the inclusion of Drake Passage data), for years 2002-2016 at each 1° x 1° grid cell. Gray

shading represents areas where the correlation does not pass significance t-tests at $p<0.05$.

Figure 5

Surface ocean $pCO_2$ trends in the SPSS biome for years 2002-2016 ($\mu$atm yr$^{-1}$): SOCATv5 data within the Drake Passage box (gray); SOCATv5 data excluding data from the Drake Passage box (green); SOCATv5 (blue); SOM-FFN product (magenta); SOM-FFN $pCO_2$ product sampled as SOCATv5 data in the Drake Passage box (light pink). Figure includes annual trends (left), austral summer trends (center) and austral winter trends (right). SOCAT-noDP winter trend omitted because it did not contain a JJA value for every year of the time-series. For reference, the atmospheric $pCO_2$ trend during the 2002-2015 period (1.95 $\mu$atm yr$^{-1}$) is shown as a horizontal black line.

Figure 6

(a) Sea-air $CO_2$ flux and (b) $pCO_2$ averaged over the Southern Ocean SPSS biome, from the SOM-FFN $pCO_2$ product (blue) and that of the SOM-FFN-noDP product created without the inclusion of Drake Passage data (red). Trends and uncertainty values in corresponding colors.

Figure 7

(a) Trajectories of Drake Passage-transiting SOCCOM floats included in this analysis. Colored diamonds represent the location of surface measurements for each float. Data from floats collected east of 55$^o$W and west of 90$^o$W are not included in this analysis. Gray dots represent observations from the DPT. (b) Mean surface ocean $pCO_2$ seasonal cycle estimate for: black: underway Drake Passage Time-series data for years 2002-2016, purple: DPT for years 2016-2017 to match years covered by the floats, and orange: SOCCOM floats. Seasonal cycles are shown on an 18-month cycle, calculated from a monthly mean time-series with the atmospheric correction to year 2017. Shading represents 1 standard error accounting for the spatial and temporal heterogeneity of the sample and the measurement error (2.7% or $\pm$11 $\mu$atm at a $pCO_2$ of 400 $\mu$atm for floats; $\pm$2 $\mu$atm for DPT data) combined using the square root of the sum of squares.

Figure 8

(a) 2002-2017 underway DPT $pCO_2$ observations (circles) and surface $pCO_2$ estimates from SOCCOM floats overlain (diamonds; $\mu$atm), plotted versus latitude. (b) Same as (a) but plotted as January 2016 to December 2017.

Figure 9

(a) Map of SOCCOM floats with DPT crossovers within 75km, 3 days, and 0.3°C SST from coincident

surface observations. (b) Calculated $pCO_2$ from SOCCOM float (x-axis) versus DPT underway $pCO_2$

observations (y-axis) for crossover float locations, with 1:1 line. Colors correspond to float number in

Figure 7. Horizontal width of shading represents SOCCOM relative standard uncertainty, which is

estimated at ±2.7% μatm; vertical shading is ±2 μatm uncertainty around DPT observations. Black 'x' and

squares indicated crossovers within a smaller window (50km/2day/0.3°C SST and 25km/1day/0.3°C SST

respectively).

Supplementary Table 1

Seasonal and annual trends as reported in Figure 5 and Supplementary Figure 3.

Supplementary Figure 1

Map of Polar-front Antarctic Zone (PAZ) which is defined at 1° x 1° resolution as the region between the

Subantarctic front and the northern extent of sea ice [Williams et al., 2017]. The red line represents the

boundaries of the location of the Subpolar seasonally stratified (SPSS) biome [Fay & McKinley, 2014].

The black box represents the Drake Passage region considered in the supplementary figures that follow.

Supplementary Figure 2

Mean surface ocean $pCO_2$ seasonal cycle estimate for years 2002-2016, for the PAZ region from each

dataset, shown on an 18-month cycle, calculated from a time-series with the atmospheric trend removed

(1.95 μatm yr$^{-1}$). Shading represents 1 standard error for biome-scale monthly means. Bar plot indicates the

number of years containing observations in a given month (maximum of 15 years) for the SOCAT-DP

SOCAT-noDP, and SOCAT-all datasets.

Supplementary Figure 3

Surface ocean $pCO_2$ trends in the PAZ region for years 2002-2016 ($\mu$atm yr$^{-1}$): SOCATv5 data within the

Drake Passage box (gray); SOCATv5 data excluding data from the Drake Passage box (green); SOCATv5

(blue); SOM-FFN product (magenta); SOM-FFN $pCO_2$ product sampled as SOCATv5 data in the Drake

Passage box (light pink). Figure includes annual trends (left), summer trends (center) and winter trends

(right). SOCAT-noDP winter trend omitted because it did not contain a JJA value for every year of the

time-series. For reference, the atmospheric $pCO_2$ trend during the 2002-2015 period (1.95 $\mu$atm yr$^{-1}$) is

shown as a horizontal black line.

**Figure 1**

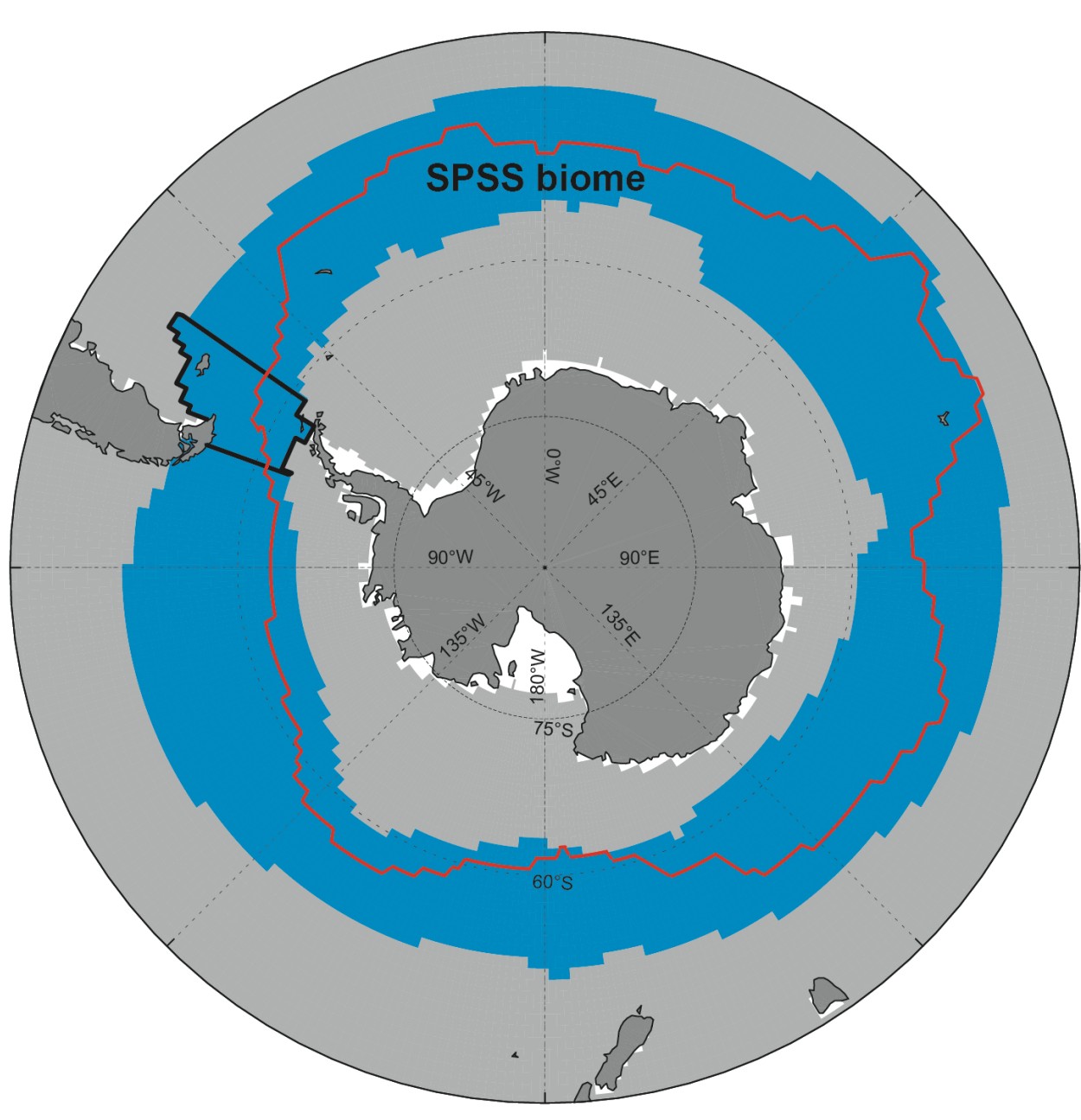

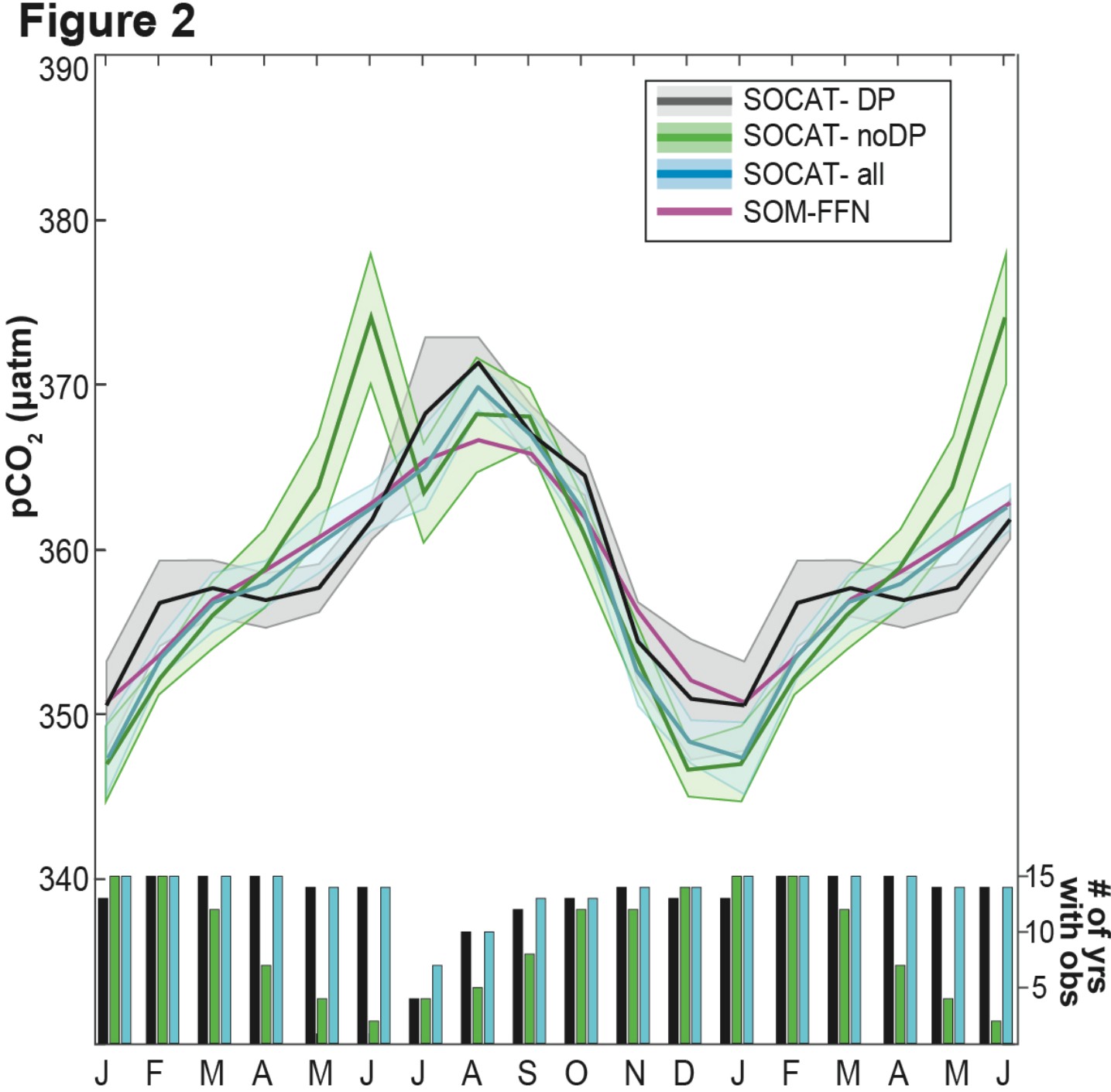

**Figure 2**

**Figure 3**

**(a)**        **Annual**             **(b)**        **JJA**

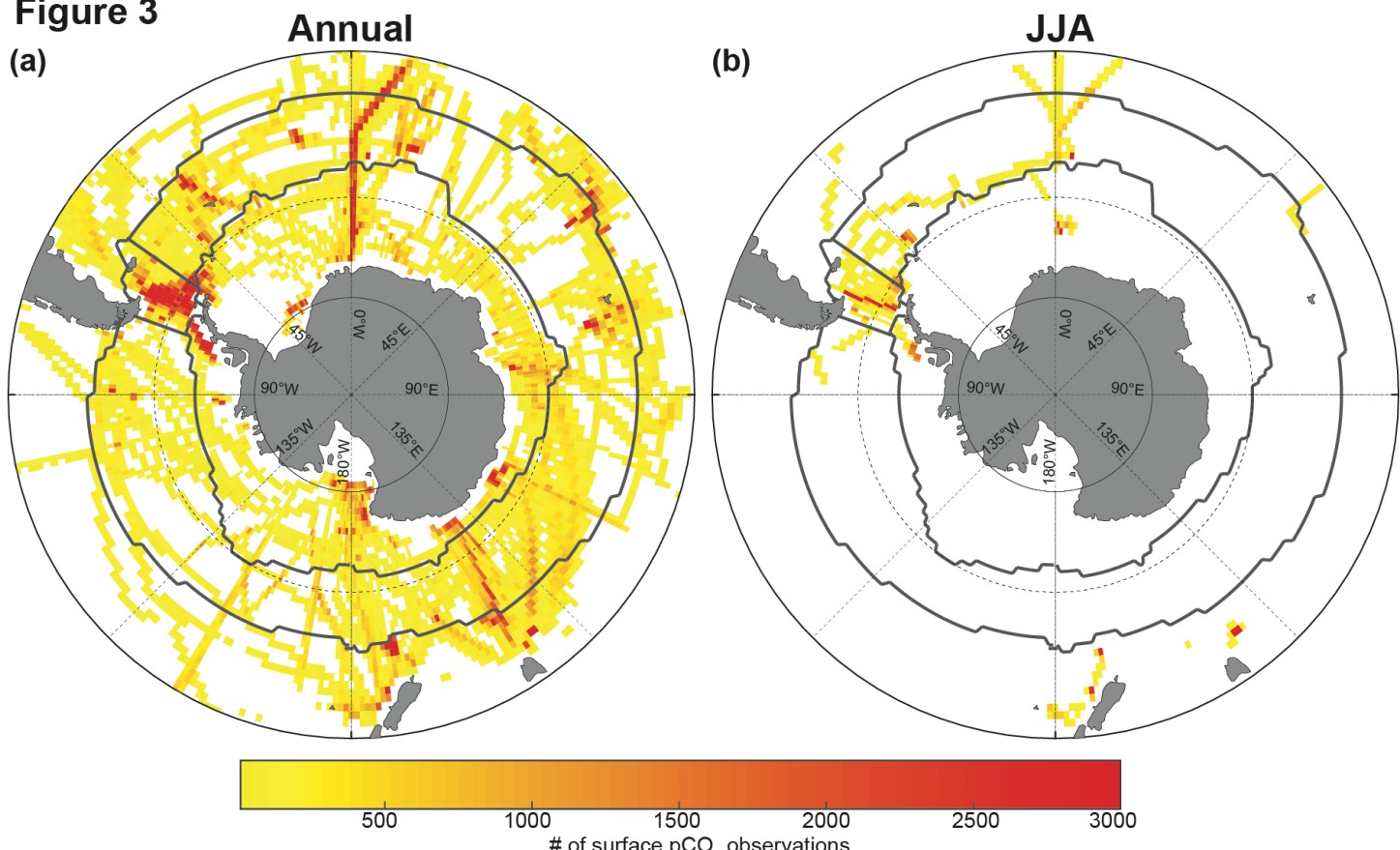

# of surface $pCO_2$ observations

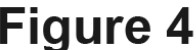

## Figure 4

**(a)**

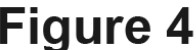

**(b)**

Correlation

**Figure 5**

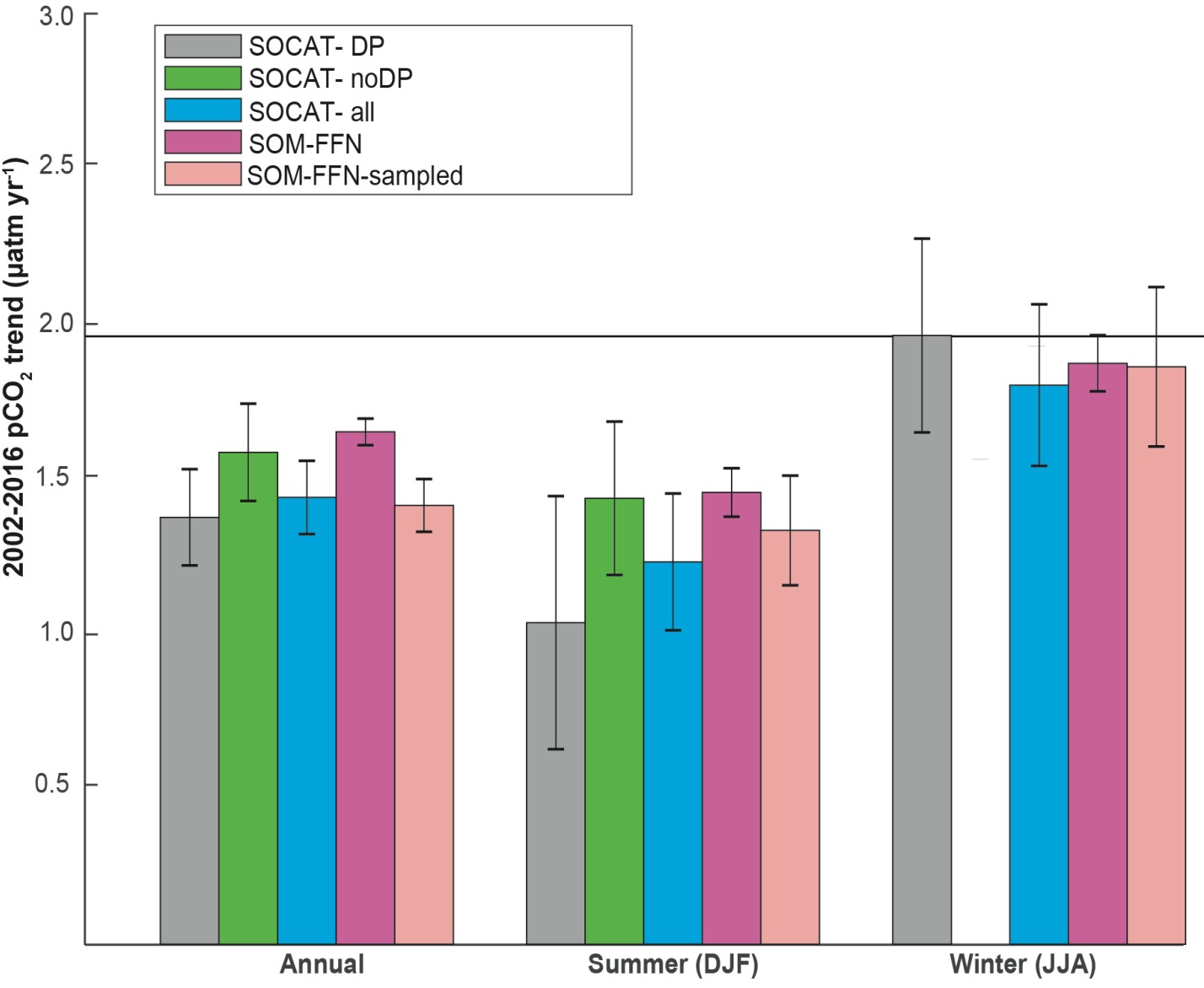

# Figure 6

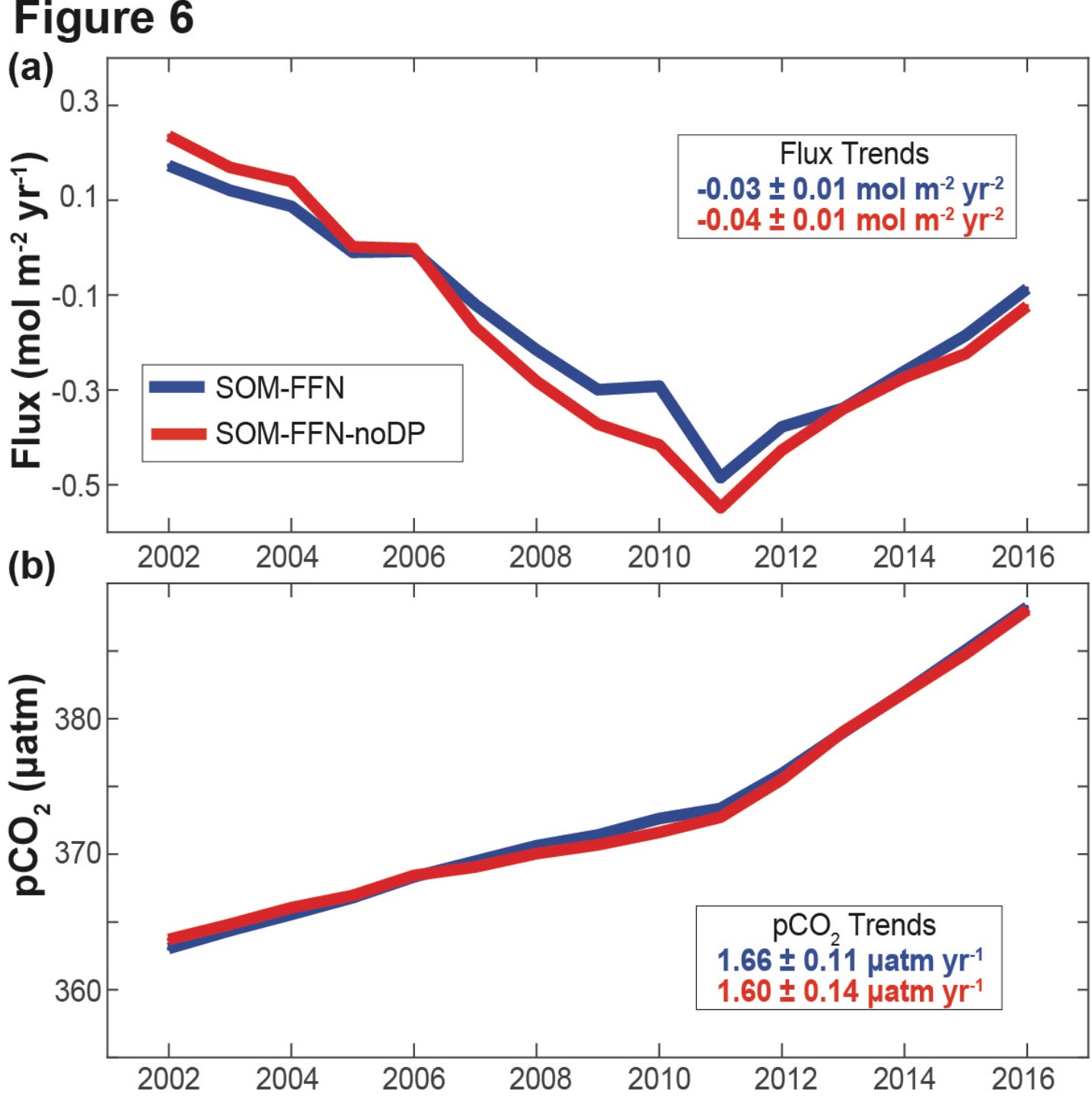

**(a)**

Flux (mol m$^{-2}$ yr$^{-1}$)

0.3

0.1

-0.1

-0.3

-0.5

Flux Trends
**-0.03 ± 0.01 mol m$^{-2}$ yr$^{-2}$**
**-0.04 ± 0.01 mol m$^{-2}$ yr$^{-2}$**

SOM-FFN
SOM-FFN-noDP

**(b)**

pCO$_2$ (µatm)

380

370

360

pCO$_2$ Trends
**1.66 ± 0.11 µatm yr$^{-1}$**
**1.60 ± 0.14 µatm yr$^{-1}$**

2002  2004  2006  2008  2010  2012  2014  2016

# Figure 7

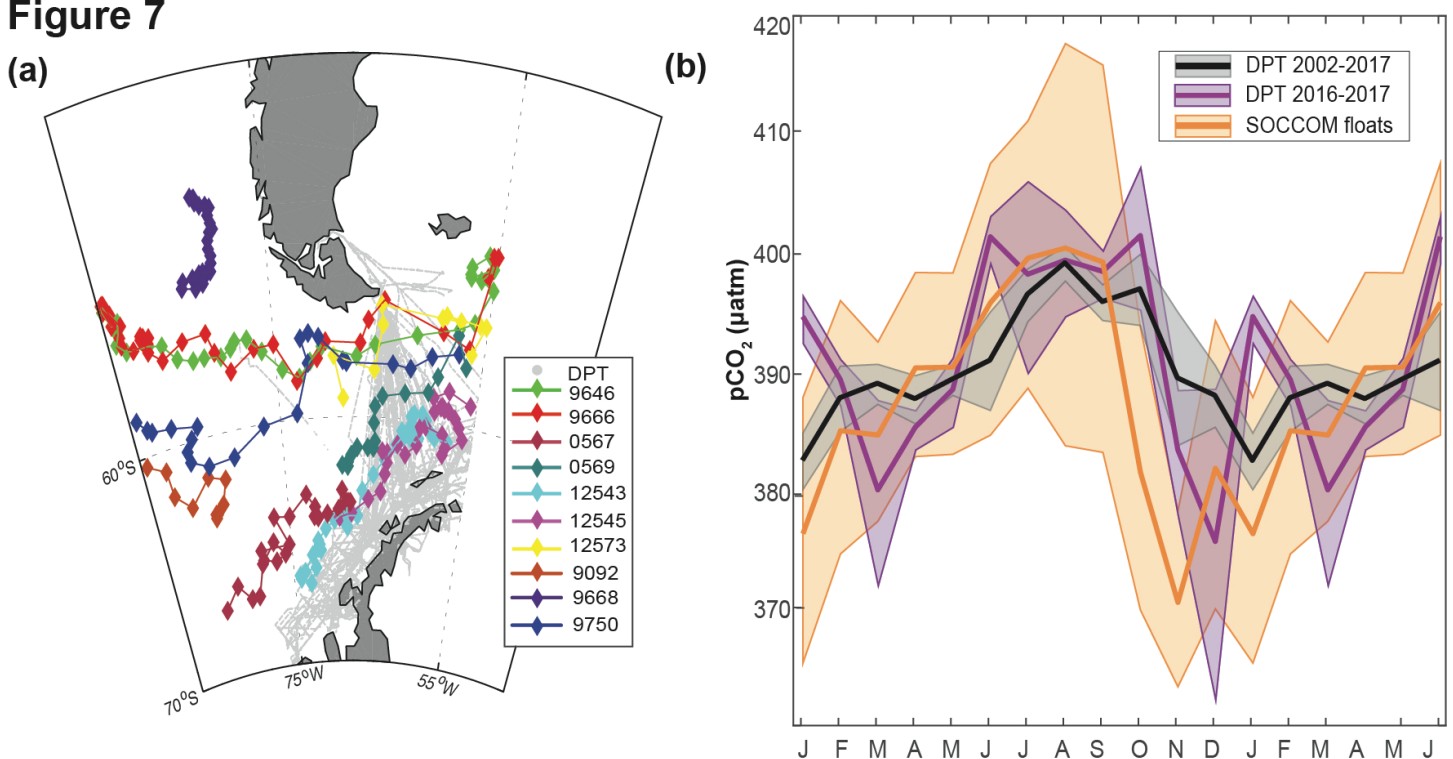

# Figure 8

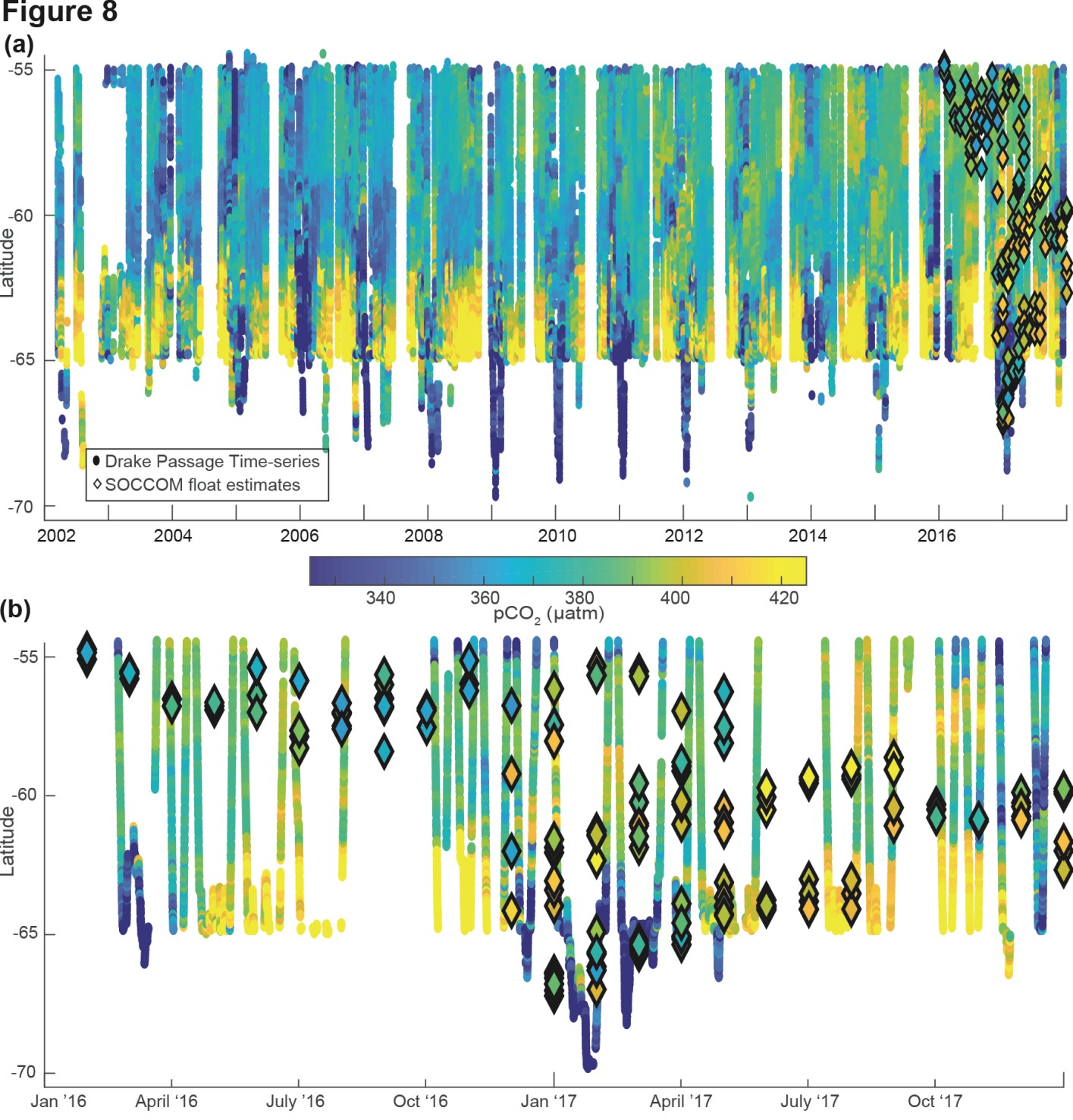

# Figure 9

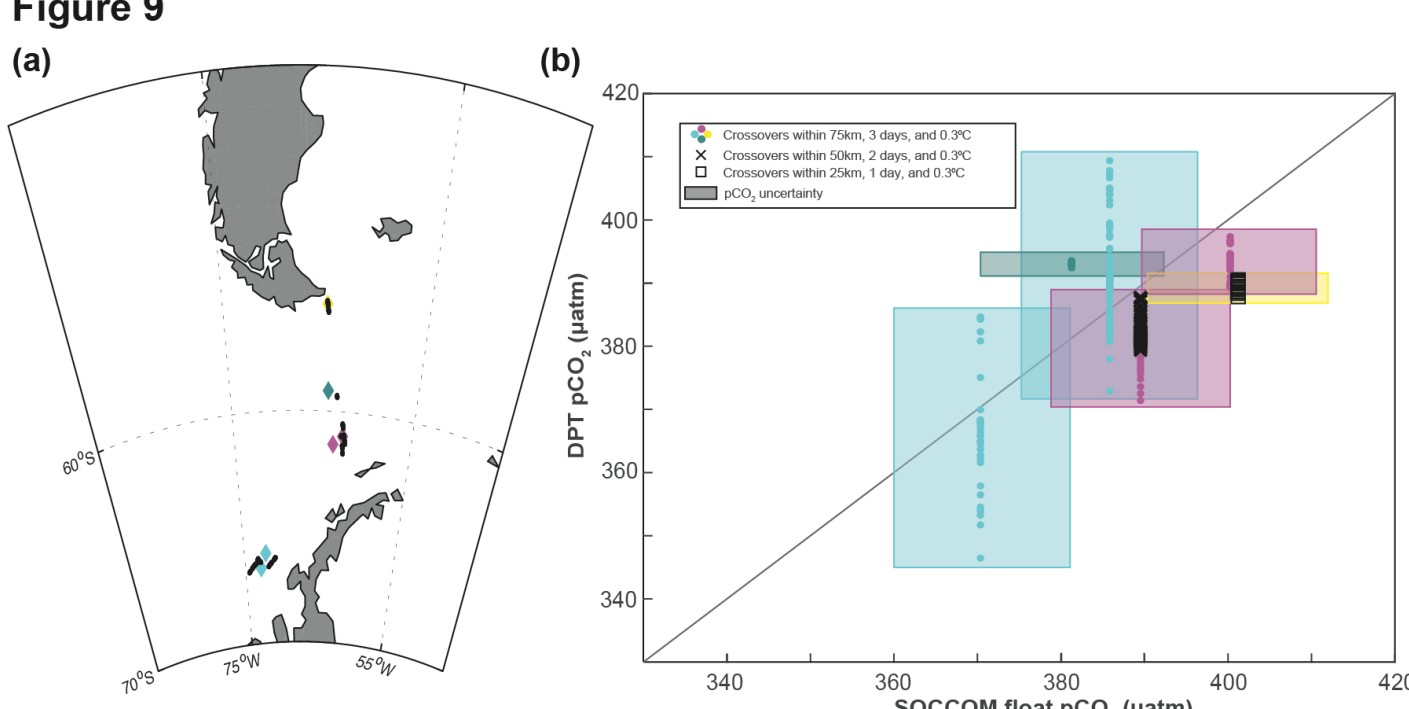

**(a)**

**(b)**