# Peer review of "Utilizing the Drake Passage Time-series to understand variability and change in subpolar Southern Ocean pCO2"

_Biogeosciences, 2017_

## Referee Comment (RC1) · Anonymous Referee #1 · 30 Dec 2017

General comment:

The Southern Ocean, south of 30S (especially in the band 35-55S) is a major ocean carbon sink, representing about half the annual global ocean carbon uptake (Takahashi et al 2012), but how this sink varies from year to year or at decadal scale is not yet well observed and understood; results also depends on methods, either based on direct observations, ocean models or atmospheric inversions (Lenton et al., 2013). Although it has been recognized that at interannual scale the variations in the tropical pacific linked to ENSO dominate the variability of the global ocean carbon sink (e.g. Rödenbeck et al, 2014), at decadal scale the Southern Ocean (SO) play an impor-

tant role (Landschützer et al., 2015). In this context it is important to better document the pCO2 spatio-temporal distribution (seasonal to decadal and long-term trends) and air-sea CO2 fluxes in the "windy" SO based on observations and/or models, such as presented in this manuscript. This is a Âń hot Âż topic that retains special attention in the community (e.g. Gregor et al., 2017; Ritter et al., 2017 for very recent analysis). In this context and following previous analyses based on Drake Passage (DP) Time-series data (Takahashi et al 2014; Munro et al 2015), that represent the most comprehensive data-set in the S.O. for all seasons and many years, authors investigate the temporal pCO2 variations observed in this region (season, interannual and trends for years 2002-2015), with the aim of comparing the results in DP with other circumpolar regions. For that authors use all pCO2 data available in SOCAT (here version 4) selected in a representative biome, here the SubPolar Seasonally Stratified (SPSS, but also sensitivity tests in PAZ) and using results from a method (SOF-FFN) that extrapolate pCO2 fields. Authors conclude that DP is broadly representative of both pCO2 seasonality and trends observed at basin scale (with some questionable remarks for winter and interannual variability). They also confirm that the ocean carbon uptake increased since 2002 in this region. This is not surprising as authors use almost the same data and method (SOF-FNN) as in previous analyses (e.g. Landschützer et al., 2015; Munro et al., 2015). Authors also investigate for the first time the pCO2 derived from SOCCOM floats in the DP and compare with DPT underway observations in 2016-2017. Their results show that pCO2 derived from floats are in the range of underway observations suggesting that data from different platforms (underway, floats) should now be mixed to better evaluate the change of the ocean carbon sink. This is particularly important as underway data in winter are very sparse in the S.O. (except in DP, as noted in this MS) and first results from SOCCOM floats suggest large CO2 source in austral winter in the polar Antarctic zone (Williams et al., 2017); such high pCO2 is also revealed in the present paper for floats in DP (June-July 2017). Merging underway and float pCO2 data is also highly relevant regarding recent discussions for future SOCAT product: should pCO2 from floats included in this data-base ? And what would be the results

and sensitivity of the methods such as SOF-FFN (or other SOCOM, Rödenbeck et al, 2015) if one uses merged data set. I think authors have in hand the data (and model) to investigate such question and this would offer a new step and deeper analysis for this manuscript.

Overall I found the analysis very attractive and I support publication after revision. My main comments/questions concern the trends results in the DP region, the SPSS biome definition and I would suggest to extend the analysis over the full observational period 2002-2017, especially given the apparent stability of the carbon sink in recent years, 2011-2015, produced by SOM-FNN and presented in this paper (but not discussed). Below I address specific and minor comments.

Specific comments:

C1: Trends: Because authors use (almost) the same data and/or method as in previous analyses (Takahashi et al., 2014; Munro et al 2015; Landschützer et al., 2015), the results on seasonality and trends (and increasing sink over 2002-2012) are not really new and should be (almost) the same as published in previous papers. However the results for the trends are surprisingly not the same. Figure 5 in the present manuscript shows trends around 1.5 $\mu$atm/yr or less depending the data and seasons. For the period 2002-2012, Takahashi et al. (2014) observed different trends in northern and southern DP sectors (Drake 1: +2.1 $\mu$atm/yr, Drake 4: 1.5 $\mu$atm/yr). These contrasting trends in the north and south of DP were later confirmed for years 2002-2015 (Munro et al., 2015), with pCO2 trends ranging from 1.74 to 1.16 $\mu$atm/yr. The annual trend presented here for DP (about 1.2-1.3 $\mu$atm/yr, Figure 5 gray) seems lower than 1.5 $\mu$atm/yr reported by Munro et al (2015) on average for DP sector. This should be discussed: is it linked to the selected zone, new data, other reasons ? I understand that when selecting data in PAZ region authors evaluate a much higher rate (as shown in Supp. Material), but this is not really discussed in the manuscript. For clarity, I would suggest to start investigating the DPT data in more detail (contrast north/south, or recall previous findings) and explain why in the following the seasonality and trends

are presented on average for the whole DP region, but limited to SPSS biome. Also, it would be useful to present in a table the values presented in Figure 5. In addition, the results presented in Figure 6 suggest significant change of the $pCO_2$ trends before and after 2011. This is not discussed in the paper and it would have been interesting to evaluate trends over 2002-2011 and 2011-2017 with the new data (2016 and 2017 are available and used to compare with floats). In addition, $pCO_2$ trend shown in Figure 6 (for S.O. south of 35S) seems low compared to SPSS (Figure 5) and PAZ (sup. Figure 3). For clarity and because the study is focus on SPSS, authors should explain why they change the boundary for the fluxes and SOM-FNN sensitivity tests (Figure 6).

C2: SPSS biome: Given the differences of the trends in the north/south DP (as well as different $pCO_2$ seasonality reported by Munro et al 2015), I was wondering if the SPSS biome definition for large-scale purpose (SST<8°C, Fay and Mc Kinley, 2014) is suitable for all data/region in DP. In the manuscript, SPSS biome is shown in a map (Figure 1), but its definition should be recalled. I also note that the original SO-SPSS biome definition used in Fay and Mc Kinley (2013) and Lovenduski et al. (2015) was different (4<SST<9°C), leading to a narrow region compared to the one used here (the DP region was included in SPSS and ICE biomes). Based on $pCO_2$ distribution would the biome criteria revised, offering a new view of biomes in the complex S.O. frontal system ?

C3: Air-sea $CO_2$ fluxes: Figure 6 shows fluxes over the period 2002-2015. The results are not new for 2002-2012 and confirm previous analysis with the same method (Landschützer, et al., 2015), i.e. an increasing sink. Interestingly, the fluxes seem relatively stable in 2011-2015 suggesting something occurred around 2011, also revealed in $pCO_2$ trends (Figure 6b). This is new but not discussed in the manuscript and should be highlighted. In addition, authors have in hand data for 2016-2017 (comparison with floats) and it would be interesting (and really new) to present and discuss results over the full observational period 2002-2017. Is the sink still increasing after 2012 ? The SOF-FFN method has been recently applied with SOCAT-V5 and results used in the

most recent global carbon budget (Le Quéré et al., 2017, cite). Why not using the most recent SOF-FNN results ? The pCO2 trends showed in Figure 6b also suggest to separate the periods 2002-2011 and 2011-2015 (or 2011-2017 if authors extend their analysis).

C4: SOCCOM floats: I understand that SOCCOM floats now offer a fantastic opportunity to compare derived pCO2 with underway data and results presented here are very encouraging. However, I find this part of the paper somehow disconnected to previous sections. Here, authors present a comparison for 2016-2017, but not discuss much about seasonality, trends and comparison with other regions (the aim of the paper). After comparing with the float data, the results should be discussed in this context. I would suggest to detail the seasonality from the floats and discuss what new information we learn compared with SOCAT and SOF-FNN (Figure 2). Given the pCO2 uncertainty (about 11 $\mu$atm for floats) are the seasonal amplitude from floats coherent with your SOCAT and SOM-FFN results ? It is not very clear from Figure 7 to see the seasonality from floats. A new figure, a plot pCO2/month like Figure 2, would be appropriate. In addition, as other floats were previously analyzed in the PAZ (Williams et al 2017) why not adding these data and compare floats in and outside the DP region (the main question addressed in the paper) ? Is the pCO2 peak in June confirmed or not with the floats ? The SOCCOM section also revealed that underway data in 2016-2017 are now available. Thus, it would be interesting to finish this paper with an analysis of the trends for the full period 2002-2017 merging pCO2 from DPT underway and floats and demonstrate why it is important to merge these data in a coherent synthesis (e.g. a next step for SOCAT ?).

Other comments and minor comments:

OC1: Line 56-64: Authors recall previous results that showed decreasing CO2 sink in the SO (80s-90s) followed with increasing sink (since 2002). Interestingly, one result presented in this paper (Figure 6a) suggest a relatively stable sink over 2011-2015. This result is new and should be discussed in the paper.

OC2: Line 110-120: DPT data. Authors list the properties measured during DPT. Not sure to understand why they refer to "TCO2, calculate nutrient and carbonate parameters", apparently not used in this manuscript. However, TCO2 data might be used with pCO2 to compare pH from SOCCOM floats (suggestion). These data might be also used to explain why authors conclude that seasonality is "controlled by a combination of temperature and deep water mixing effects". However, process analysis is not presented in this study.

OC3: Line 121-129: SOCAT data: Authors should specify in more detail the data they use; are they use all original SOCAT data (WOCE flag 2) with cruise flags A,B,C,D,E or only A,B,C,D (i.e. not include data from buoys, see also comment below for Figure 3) ? SOCAT-V4 is used here, but SOCAT-V5 was made public in June. Would it be possible to extend your analysis with SOCAT-V5 (SOM-FFN product should be available with SOCAT-V5, Le Quéré et al, 2017). Although it is specified later (line 190), authors should indicate in this section that they construct their own monthly 1x1 fields based on SOCAT original data (i.e. they not use SOCAT gridded products).

OC4: Line 123: Detail: Data in 1957-1958 were first included and QCed in SOCAT-V3 (Bakker et al 2016).

OC5: Line 130-144: SOM-FFN: Could you specify how the data are extracted from SOM-FFN for the SPSS biome: is it based on SST criteria (and from SOM-FFN SST) or Lat/Long grid ?

OC6: Line 150: pCO2 from floats are derived from pH and reconstructed alkalinity. As this study is focus on DPT data, it would be interesting to show first the pH cycles (measured by floats) and compare with pH data calculated from pCO2/TCO2 DPT data. Also, how well TA is reproduced in this region (either based on TA/SSS/SST relations or TCO2/pCO2 data ?). Recall however that uncertainty on derived pCO2 is not very sensitive to TA (Williams et al., 2017).

OC7: Line 150: Johnson and Claustre, (2016) not in reference

OC8: Line 157: Authors have now in hand data to explore if the merge of pCO2 underway and from floats would be useful to include in SOCAT (a question now regularly discussed in SOCAT community). Why not merging these data now and re-evaluate trends over 2002-2017 (with or without floats) ? The results would be really supportive to progress on international data synthesis.

OC9: Line 170: SPSS: the analysis is strongly based on this biome. The biome definition should be recalled (is it SST<8°C or 4>SST>9°C ?); is this criteria suitable for all seasons and all waters in DP ? For the interannual pCO2 variability analysis, is the SPSS biome boundary used here is climatological or interannual (Fay and Mc Kinley, 2014) ? Maybe interannual variability of the biome is minor for this analysis.

OC10: Line 181: SPSS versus PAZ: authors indicate that conclusions remain unchanged. However, pCO2 trends seem significantly different especially in austral winter (Figure 5 and sup. Figure 3). This should be discussed in section "'4.3 Trends".

OC11: Line 210: Seasonal cycle: mean seasonal pCO2 amplitude in the SPSS is 23 $\mu$atm. This is very coherent with previous studies including low amplitude compared to north SPSS (e.g. see SO-SPSS results for all method in SOCOM, Rödenbeck et al, 2015). But interestingly, amplitude of pCO2 seasonality derived from floats in the Polar Antarctic Zone seems much larger (Williams et al., 2017) with some floats showing winter pCO2 much higher than in the atmosphere, a signal apparently not revealed in SOCAT-DP, SOCAT- noDP, or SOM-FFN (Figure 2). On the other hand, in the DP, Munro et al (2015) reported seasonal cycles significantly different in the north and south (large cycle in the north, small cycle in the south). These results should be included in the discussion.

OC12: Line 216-217: If processes are not evaluated here (SST versus TCO2), add a reference (e.g. Munro et al., 2015).

OC13: Line 218-220: For clarity, specify that original xCO2 (in ppm) data for atmospheric concentrations are converted to pCO2 (in $\mu$atm). Are the atmospheric values

represent global mean or only southern hemisphere ?

OC14: Line 221: Results and discussion on uncertainty of the seasonal mean are not clear (for me). Compared to measurements accuracy (2-5 $\mu$atm) these uncertainties seem small (5 $\mu$atm or less) suggesting that seasonal cycle is relatively well constrained with available observations. Correct ? OC15: Line 227: What is the origin of the pCO2 peak in June (in both SPSS and PAZ) ? An anomaly related to a specific location, cruises, year ? Is such signal also seen in floats ? Why this is not reproduced in SOM-FNN ?

OC16: Line 230: Figure 3: I am surprised some data in SOCAT are not (apparently) seen in austral winter (JJA). Maybe change the color scale to better highlight the locations of the data (white-yellow not very clear on this figure). I would suspect authors used SOCAT-V4 for Cruises with Flag A, B, C, D, i.e. pCO2 data from drifting buoys (flag E) not included. (however, later in this paper authors use pCO2 derived from pH SOCCOM floats). If this is correct, authors should specify why they are not using all SOCAT-V4 data, including pCO2 from 14 drifting buoys in the S.O. in 2002-2011 (and one launched in DP).

OC17: Line 255: For the anomaly in June could you explain why you "expect data to be less similar to that collected in the Drake Passage."

OC18: Line 265: Authors indicate that "the Drake Passage seasonal cycle is representative of the broader SPSS biome seasonality, based on the available observations to date." However recent data from floats suggest high pCO2 in winter, well above 400 $\mu$atm (Williams et al 2017) and one would conclude that DP is not always representative of the SPSS.

OC19: Line 267: Again, in May-June, some data from buoys are available in SOCAT (values ranging between 320 and 400 $\mu$atm for these months). Are these data used in your study ?

OC20: Line 270: Authors suggest that SOM-FNN is likely being driven by Drake Passage data. However in Figure 6 they show that SOM-FNN with DP or no-DP lead to the about same results. It would be interesting to show the seasonality derived from SOM-FNN with SOCAT-noDP (like for the trends presented in Figure 6).

OC21: Line 273: To conclude on the seasonality, because authors construct a new "climatology" for SPSS based on data for the period 2002-2015, it would have been interesting to compare and discuss these cycles (DP and no-DP) with previous climatologies (e.g. Takahashi et al 2009, 2014) that used data back to the 80s (including winter cruises in the 90s in the SPSS but not used in the present study).

OC22: Line 286: Could you really compare your results with the model of Lovenduski et al (2015) ? The model was applied for a different period (1981-2007) and the SPSS biome in this model seems different (back to my comment on SPSS definition). To highlight the low IAV in the model, would it be possible to plot the model results in DP like for observations (Figure 4a) ? These results suggest that the model is not able to capture correct IAV and this is important to notice as such models are also used for prediction. Is your analysis in DP would help to identify processes that should be first revised in the model (in few words: dynamics, mixing, biology or others) ?

OC23: Line 292: To better follow the discussion on IAV, would be nice to add a figure, like Figure 4a but for SOM-FFN in DP. How your IAV results impact on the next steps, i.e. errors associated to long-term trends and fluxes analyses (Figure 6) and our understanding of the S.O. carbon sink variability ?

OC24: Line 325-330: All trends presented in Figure 5 are below atmospheric value. However, in previous analysis (and in the PAZ, Supp Figure 3) trends were near or above atmospheric level in north DP (Takahashi et al 2014; Munro et al 2015). These differences should be discussed (related to new data, region selected, years ?). Results presented in Figure 5 should be listed in a table (or Supp. Mat.). Another conclusion: the trends, about 1.5 $\mu$atm/yr, confirm the corrections applied in pCO2 climatologies for reference year (Takhashi et al., 2009; 2014).

OC25: Line 350-355: Results presented in Figure 6 lead to several questions. First, as previous sections focused on SPSS biome, why Figure 6 now shows results for the Southern Ocean, south of 35°S ? Are the results for SPSS only lead to the same conclusions ? Second, it is interesting that results of the SOM-FNN with or without DP data lead to the same conclusion: the sink increases. Why the flux (and pCO2 trends) are so close with or without DP data ? Would that means that no-DP data are suitable to reproduce the increasing sink ? Or is it because results are shown for all regions south of 35°S (i.e. more data in the band 35-50°S) ? It would be interesting to show a map of the pCO2 trends from both SOM-FNN with and without DP data to identify the main differences (if any). On the other hand, the results suggest stability in 2011-2015. I think this is new and should be discussed. What is the origin of the shift around 2011 ? This seems related to no-DP data (correct ?).

OC26: Line 370: Taking into account all uncertainties (standards, SST and equilibrium temperature calibrations,...) the accuracy of +/-2 $\mu$atm for ship-based systems is the best that can be achieved. Many cruises in SOCAT received a flag (C,D) with accuracy of +/-5 $\mu$atm (about half the floats, 11 $\mu$atm).

OC27: Line 371: "there is great potential for these two observational platforms to work in concert". Yes, I fully agree. This could be done in this study, e.g. exploring the seasonality with merging product, or extend trend analysis to 2017.

OC28: Line 374: If you use SOCAT-V5, other data are available in 2016. Might be useful to explore and add few more cross-overs with floats.

OC29: Line 375-380: Figure 7 shows nicely when and where floats data are available, but comparison of underway pCO2 and floats is not easy to see on figure 7. Authors should add a plot of pCO2 versus time for the period 2016-2017 (e.g. like figure 5 in Williams et al., 2017).

OC30: Line 380: pCO2 data from floats in June-July 2017 suggest high pCO2. What are the values (not clear in figure 7) ? Are these values coherent with other floats in the SPSS in other sectors (float 9096, Williams et al., 2017).

OC31: Line 406-420: Authors recalled that pCO2 derived from floats are subject to uncertainties associated to pH calibration, TA reconstruction, etc.. Here, they have in hand TCO2 and pCO2 data from DPT, and it would be interesting to quantify comparison for Salinity (used for TA), TA and pH (from DPT TCO2/pCO2 data). Would this leads to the same accuracy (+/- 11 $\mu$atm). What TA algorithm is used in your calculations ? Would it be appropriate to use an algorithm specific of the DP region ?

OC32: Line 415: Authors recalled that in previous analysis, mean difference with ship data was only 3.7 $\mu$atm. What mean difference did you get in your comparison at DP ?

OC33: Line 426-427: Authors conclude: "With this complete coverage we find seasonal amplitudes in the SPSS to be smaller than subpolar regions in the Northern Hemisphere, and controlled by a combination of temperature and deep water mixing effects". I think this is not really new but confirm previous results (e.g. Takahashi et al 2009; Rödenbeck et al., 2015) and authors did not evaluate processes (temperature, mixing) in their study.

OC34: Line 426-427: Authors conclude: "Uncertainties in the seasonality remain considerable". What is "considerable"? Would the conclusion the same if all SOCAT data are used, including data back in the 90s with more cruises conducted in winter ?

OC35: Line 430: do we need such analysis to conclude that there is a lack of winter data ? This is an important message for future observations, but for the past decades the only way to mimic winter data is to reconstruct at best pCO2 fields (such as SOM-FNN, or ocean models). For the Southern Ocean, I understand that the CO2 sink and its decadal variability is relatively well-known (e.g. Landschützer et al., 2015 and your figure 6) although winter data were sparse.

OC36: Line 445-447: Authors conclude: "Southern Ocean has been a growing sink for atmospheric carbon since 2002." Again, from your results (Figure 6) it appears that ocean pCO2 increased faster in recent years (2011-2015) compared to 2002-2011, and fluxes relatively stable. This result, not discussed in the manuscript, argues to maintain long-time series observations to better detect how (and why) the ocean pCO2 is changing, and subsequently the fluxes in this important region.

OC37: Line 449: Authors conclude: "Comparisons between underway DPT and SOC-COM float measurements show general agreement". What is "general agreement" ? Please give a number (e.g. mean differences). Are your comparisons, here specifically for the DP region, confirm or not previous analyses (difference around 4 $\mu$atm, Williams et al., 2017) ?

OC38: Line 452: Authors conclude: '...could aid in reducing the uncertainty on the float pCO2 measurements by helping to identify problematic float sensors.". In your analysis, did you experienced problem with sensors ?

OC39: Line 454 : Authors conclude that a coordinated monitoring efforts that combine underway and float data is highly needed. I totally agree. If achieved, this would be a very important step, not only for the Southern Ocean as a test experience, but should be engaged at global scale (a new dream). In this context, and authors have the data in hand, I think the present study should present results combining underway and floats data (at least for the seasonality in DP, maybe for the trends 2002-2017 as a first test).

OC40: Line 535: Correct reference: Gruber et al 2009

OC41: Line 700: Correct reference: Williams et al 2017l

OC42: Figure 1: add name of other colored biomes (ICE) ? This map might be extended to 35S (the limit chosen for figure 6) ?

OC43: Figure 3: Legend: add that gray lines depict the SPPSS biome. Change color-scale (white-yellow not very clear).

OC44: Figure 7: legend 7c: the plot starts from Jan-2016 (not Oct 2015)

;;;;;;;;;;;; References added in this review (not listed in the manuscript):

Gregor, L., Kok, S., and Monteiro, P. M. S., 2017. Empirical methods for the estimation of Southern Ocean CO2: support vector and random forest regression, Biogeosciences, 14, 5551-5569, https://doi.org/10.5194/bg-14-5551-2017.

Lenton, A., B. Tilbrook, R. M. Law, D. Bakker, S. C. Doney, N. Gruber, M.Ishii, M. Hoppema, N. S. Lovenduski, R. J. Matear, B. I. McNeil, N. Metzl, S. E. Mikaloff Fletcher, P. M. S. Monteiro, C. Rödenbeck, C. Sweeney, and T. Takahashi. Sea-Âňair CO2 fluxes in the Southern Ocean for the period 1990-Âň2009, 2013. Biogeosciences, 10, 4037-4054, doi:10.5194/bg-10-4037-2013.

Ritter, R., P. Landschützer, N. Gruber, A. R. Fay, Y. Iida, S. Jones, S. Nakaoka, G. –H. Park, P. Peylin, C. Rödenbeck, K. B. Rodgers, J. D. Shutler, and J. Zeng, 2017. Observation-based Trends of the Southern Ocean Carbon Sink, Geophys. Res. Lett., 44,doi:10.1002/2017GL074837.

Rödenbeck, C., D. C. E. Bakker, N. Metzl, A. Olsen, C. Sabine, N. Cassar, F. Reum, R. F. Keeling, and M. Heimann, 2014. Interannual sea–air CO2 flux variability from an observation-driven ocean mixed-layer scheme. Biogeosciences, 11, 4599-4613, 2014 doi:10.5194/bg-11-4599-2014.

Takahashi, T., S.C. Sutherland, D.W. Chipman, J.G. Goddard, Cheng Ho, T. Newberger, C. Sweeney, D.R. Munro, 2014. Climatological Distributions of pH, pCO2, Total CO2, Alkalinity, and CaCO3 Saturation in the Global Surface Ocean, and Temporal Changes at Selected Locations. Marine Chemistry. doi: 10.1016/j.jmarchem.2014.06.004.

End review

---

## Referee Comment (RC2) · Anonymous Referee #2 · 7 Apr 2018

The authors report a thorough analysis of Southern Ocean pCO2 dynamics. This is an important topic because the Southern Ocean is a major global carbon sink and its seasonality and long-term evolution need to be further constrained. The authors focus on the Drake Passage because of its high data coverage and compare Drake Passage time-series data with data products SOCAT version 4 and SOM-FFN. They also compare Drake Passage time-series data with less precise float data from the recent and ongoing SOCCOM program.

The manuscript is very well written with easy to follow logic and I therefore recommend publication pending minor revision. My main point is that the SOCCOM data

are somewhat underexplored in this paper. I recommend the authors to provide some more details to the reader about SOCCOM data quality and to present a more detailed comparison of SOCCOM and Drake Passage time-series data for the period of overlap. Taking the two data sets independently would one extract similar seasonality patterns?

---

## Author Comment (AC1) · 11 May 2018

Review 1 Response

Reviewer 1 had three main comments regarding the analysis completed in this manuscript. They include the trend results reported for the Drake Passage region, the definition of the Subpolar Seasonally Stratified biome, and lastly the time period considered for analysis. We thank the reviewer for their thorough response to our manuscript and will respond to each of their comments and concerns below.

C1: The first specific comment refers to the trends reported in the manuscript and the "newness" of these results. The reviewer refers to previous analyses including Takahashi et al. 2014, Munro et al. 2015, and Landschützer et al. 2015 and questions why our results are different and how they represent a new analysis. We acknowledge that the trends for the SPSS biome as a whole are not groundbreaking results (similar analysis done in Fay and McKinley, 2013 and further discussed in Fay et al. 2014), but remind the reviewer that the purpose of our manuscript is the (as yet unexplored) comparison of the larger biome to the Drake Passage region. In fact, each of the papers referenced by the reviewer uses a different data set for analysis: Takahashi et al. 2014 uses the LDEO database, Munro et al. 2015 uses underway Drake Passage data, and Landschützer et al. 2015 results are based on a neural network interpolation method. Besides these differences in the datasets, small differences in analysis method can significantly influence the resulting trends, specifically for the Southern Ocean (Fay et al. 2014).

The reviewer highlights the difference between our reported trends for SOCAT-DP and the findings reported in Munro et al. 2015. The trends are not statistically different given their reported uncertainties (1.52 +/- .15 µatm/yr in Munro et al. 2015 and 1.33 +/- .16 µatm/yr in this analysis for years 2002-2015). Many differences exist in the analysis methodology, which can account for these differences. Munro et al. average underway Drake Passage data spatially to the four defined regions sitting parallel to the mean flow of the ACC. In this analysis, we use data from the SOCAT dataset (which has its own quality control methods and does not necessarily include all underway Drake Passage data) and average spatially to 1°x1° grid cells. Text was added to the manuscript to elucidate these differences. The reviewer's suggestion to include a table of trends shown in Figure 5 is much appreciated and is now included as a supplementary table.

The reviewer suggested also that we further discuss the differences between the northern and southern portions of the Drake Passage region. We did analyze our trends north and south of the front and find results that agree with conclusions presented in other research (Munro et al. 2015) that the northern section reports higher trends than the southern portion and that the seasonal cycle south of the front is significantly different from that in the north. Munro et al. 2015 discusses the reasons behind this difference and therefore we did not find it necessary to investigate in more detail. We have added to the text to recall these previous findings more clearly, and note that our overall results do not change when looking at annual trends for SOCAT-DP and SOCAT-noDP: North of the front, trends for SOCAT-DP and SOCAT-noDP are not statistically different while South of the front we see lower trends overall as seen by others, but again, the SOCAT-DP and SOCAT-noDP trends are not statistically different from one another. Therefore, our conclusions regarding the representatitvity of the Drake Passage to the larger subpolar region remain unchanged.

The last portion of this comment refers to Figure 6 and the change seen in the flux and $pCO_2$ trends around 2011. We agree that this is an interesting finding, which has also been highlighted in Gregor et al. 2018. Additionally, other efforts are currently underway that focus explicitly on the recent trend stagnation/reversal, its spatial structure and its drivers (e.g. Keppler and Landschützer, in prep). We disagree however that comparing subtrends for 2002-2011 and 2011-2015 in this study would be a valuable addition as the second time period is quite short and as previous analysis has shown, trends over short time series are highly influenced by the specific start and end year and potential anomalies that occur in those years, and are often not indicative of a larger overall shift. It is outside of the scope of this paper to investigate this potential shift in 2011, but it would be a potential next step for research building off of this paper.

We do however agree that it could be confusing to the reader that for Figure 6 we utilize a different region for the SOM-FFN sensitivity test. We have remade this figure using the SPSS biome as the region. While the axes of the figure are different because of a mean offset of the flux and pCO₂ found in this more southern region, the trends remain indistinguishable between the SOM-FFN and SOM-FFN-noDP. We have replaced Figure 6 in the manuscript.

Luke Gregor, Schalk Kok, and Pedro M. S. Monteiro, Biogeosciences, 15, 2361 2378, https://doi.org/10.5194/bg-15-2361-2018, 2018).

**Figure 6**

[Figure]

C2: The reviewer suggests including the definition of the SPSS biome in addition to referencing Fay & McKinley, 2014. We have added this definition. The SPSS biome covers nearly all of the region between the tip of South America and the Antarctic Peninsula, which is not influenced by ice. Additionally, for this analysis, we eliminate data within 50km of land (as defined by the SOCAT distance to land reported value) so as to further ensure we are not including data that could be influenced by runoff and ice melt. As pointed out in the reviewer comment, the region in

the SPSS biome which lies north and south of the Antarctic Polar Front does exhibit a gradient in $pCO_2$ (Munro et al. 2015). We account for the possibility of background gradients influencing trends due to sparse sampling over the large biomes by removing a background climatology before averaging up to biome scale. We refer the reviewer to McKinley et al. 2011 and Fay & McKinley, 2013 for further details of our methodology.

C3: As discussed in response to C1, we have remade Figure 6 to focus on the SPSS biome region, but also extended the timeseries through 2016 with an updated SOM-FFN product. We thank the reviewer for this suggestion. We agree that there are interesting shifts evident in the time series, however that analysis is not in the scope of this paper and not related to the purpose of Figure 6 as it relates to this work. We are aware of others working towards understanding this shift and we are interested to see if it continues to appear as a persistent signal as additional years of observations allow the product to be extended. We also suggest the recent publication by Gregor et al. 2018 as it discusses drivers of the Southern Ocean using the SOM-FFN as well as two additional products over a similar time period.

Additionally, the reviewer's comments that we have data through 2017 are partially incorrect. SOCATv5, which this manuscript has been updated with, includes data through December 2016 with only few data in January 2017. The portion of this study that shows data for 2017 is simply the comparison between the underway Drake Passage data and the SOCCOM floats. Additionally, as it relates to the SOM-FFN product, the most recent update extends to December 2016 as it is only based on years with complete SOCAT data coverage and therefore cannot be extended any further until the next SOCAT release (SOCATv6, expected summer 2018).

C4: We thank the author for their opinion on how the SOCCOM float comparisons seem disjointed from the previous portion of the manuscript. We have worked on the transitions to make this less abrupt. Our analysis of trends and seasonality show that there is no statistically significant difference between the Drake Passage region and the larger subpolar Southern Ocean, given the ship-based data available. We see this next section as a lead-in to future work that has the potential to increase the available observations in this sparsely sampled region, ultimately potentially lowering our uncertainty on these trends and seasonal cycles.

While the reviewer brings up many great questions and suggestions regarding how to incorporate SOCCOM float data and how it can lead to new understandings when compared with SOCAT and SOM-FFN, we direct them to forthcoming publications (Gray et al. in review, Gray et al. in prep.) which focus on these questions in much more detail. Our preliminary fine-scale comparison of the SOCCOM floats to underway Drake Passage data (Figure 8) provides an important measure of confidence for possible future efforts to incorporate float data into the SOCAT database. Much work is currently being done on this topic and it is an exciting emerging topic for the ocean carbon community.

Other comments and minor comments:

OC1: Line 56-64: Authors recall previous results that showed decreasing $CO_2$ sink in the SO (80s-90s) followed with increasing sink (since 2002). Interestingly, one result presented in this paper (Figure 6a) suggests a relatively stable sink over 2011-2015. This result is new and should be discussed in the paper. We thank the author for their suggestion but feel that a thorough discussion of this is outside the scope of this paper. Additionally, these findings are not entirely new as they have been shown in Gregor et al., 2018 who compare the sea surface $pCO_2$ and its drivers across various products.

OC2: Line 110-120: DPT data. Authors list the properties measured during DPT. Not sure to understand why they refer to "$TCO_2$, calculate nutrient and carbonate parameters", apparently not used in this manuscript. However, $TCO_2$ data might be used with $pCO_2$ to compare pH from SOCCOM floats (suggestion). These data might be also used to explain why authors conclude

that seasonality is "controlled by a combination of temperature and deep water mixing effects". However, process analysis is not presented in this study. We agree that it may cause confusion that we mention additional parameters measured as part of the Drake Passage time series but do not discuss or utilize them in our analysis. We have revised this paragraph in an attempt to clarify this. The other suggestions are reiterated in further reviewer comments and will be responded to accordingly.

OC3: Line 121-129: SOCAT data: Authors should specify in more detail the data they use; are they use all original SOCAT data (WOCE flag 2) with cruise flags A,B,C,D,E or only A,B,C,D (i.e. not include data from buoys, see also comment below for Figure 3) ? SOCAT-V4 is used here, but SOCAT-V5 was made public in June. Would it be possible to extend your analysis with SOCAT-V5 (SOM-FFN product should be available with SOCAT-V5, Le Quéré et al, 2017). Although it is specified later (line 190), authors should indicate in this section that they construct their own monthly 1x1 fields based on SOCAT original data (i.e. they not use SOCAT gridded products). We have updated our analysis to SOCATv5. We have added the specific WOCE and cruise flags included in our analysis. We feel that the explanation of our monthly 1°x1° calculations is best suited in the methods section and restrict the dataset discussion to an overall description of the actual data that is included.

OC4: Line 123: Detail: Data in 1957-1958 were first included and QCed in SOCAT-V3 (Bakker et al 2016). We thank the reviewer for this clarification. We have reworded this paragraph and this portion of the statement was removed for clarity.

OC5: Line 130-144: SOM-FFN: Could you specify how the data are extracted from SOM-FFN for the SPSS biome: is it based on SST criteria (and from SOM-FFN SST) or Lat/Long grid ? The data are extracted based on lat/lon criteria.

OC6: Line 150: $pCO_2$ from floats are derived from pH and reconstructed alkalinity. As this study is focus on DPT data, it would be interesting to show first the pH cycles (measured by floats) and compare with pH data calculated from $pCO_2/TCO_2$ DPT data. Also, how well TA is reproduced in this region (either based on TA/SSS/SST relations or $TCO_2/pCO_2$ data ?). Recall however that uncertainty on derived $pCO_2$ is not very sensitive to TA (Williams et al., 2017). We thank the reviewer for this suggestion but feel it is outside of the scope of this analysis as the comparison between floats and the DPT is preliminary and there is much potential future work that would constitute an additional manuscript.

OC7: Line 150: Johnson and Claustre, (2016) not in reference. Thank you for pointing out this omission. It has been added to the reference list.

OC8: Line 157: Authors have now in hand data to explore if the merge of $pCO_2$ underway and from floats would be useful to include in SOCAT (a question now regularly discussed in SOCAT community). Why not merging these data now and re-evaluate trends over 2002-2017 (with or without floats)? The results would be really supportive to progress on international data synthesis. We thank the reviewer for this suggestion but feel it is outside of the scope of this analysis. Other groups are currently working on this topic using a variety of approaches (Gray et al. in review, Gray et al. in prep.).

OC9: Line 170: SPSS: the analysis is strongly based on this biome. The biome definition should be recalled (is it SST<8$^o$C or 4>SST>9$^o$C ?); is this criteria suitable for all seasons and all waters in DP? For the interannual $pCO_2$ variability analysis, is the SPSS biome boundary used here is climatological or interannual (Fay and McKinley, 2014)? Maybe interannual variability of the biome is minor for this analysis. We first direct to our response to the reviewer's second major point (C2) where we address the biome definition. We have added the definition to the manuscript. In regard to the question of if this criteria is suitable for "all waters in DP", the point of the biome definition is not to encompass the DP but to define waters that exhibit subpolar traits. Further explanation of the biome assignments and history of definition is available in Fay &

McKinley, 2014. The biomes used here are climatological. Fay & McKinley 2014 also show how biome area changes as a function of year if interannual biomes are considered and their findings show that the Southern Ocean SPSS biome area does not vary much for years 1998-2010.

OC10: Line 181: SPSS versus PAZ: authors indicate that conclusions remain unchanged. However, $pCO_2$ trends seem significantly different especially in austral winter (Figure 5 and sup. Figure 3). This should be discussed in section "'4.3 Trends". We thank the reviewer for this suggestion. This portion of the manuscript has changed during the revision process but we acknowledge that the results shown initially did produce different results in the austral winter. Our statement about conclusions remaining unchanged referred to the trends all being indistinguishable when using the "DP", "no-DP", or "all" datasets (gray, green, and blue bars in each of the respective figures). The fact that trends are different for the PAZ versus the SPSS biome is understandable given the different areas they cover. Specifically looking at the Drake Passage itself, the PAZ has a much more limited spatial area in that region than that encompassed by the SPSS biome. We have elaborated on this comparison in our revised version section 4.3 as suggested.

OC11: Line 210: Seasonal cycle: mean seasonal $pCO_2$ amplitude in the SPSS is 23 µatm. This is very coherent with previous studies including low amplitude compared to north SPSS (e.g. see SO-SPSS results for all method in SOCOM, Rödenbeck et al, 2015). But interestingly, amplitude of $pCO_2$ seasonality derived from floats in the Polar Antarctic Zone seems much larger (Williams et al., 2017) with some floats showing winter $pCO_2$ much higher than in the atmosphere, a signal apparently not revealed in SOCAT-DP, SOCAT- noDP, or SOM-FFN (Figure 2). On the other hand, in the DP, Munro et al (2015) reported seasonal cycles significantly different in the north and south (large cycle in the north, small cycle in the south). These results should be included in the discussion. We have added additional discussion of these comparison studies. We would like to point out however that the referenced comparison to Williams et al. 2017 for amplitudes derived from floats in the PAZ is in fact only based on one float which has one complete seasonal cycle as shown in Figure 5c of Williams et al. 2017. Therefore it would not be unexpected for one occurrence to differ from a mean or climatological value based on decades of data.

OC12: Line 216-217: If processes are not evaluated here (SST versus $TCO_2$), add a reference (e.g. Munro et al., 2015). Reference has been added.

OC13: Line 218-220: For clarity, specify that original $xCO_2$ (in ppm) data for atmospheric concentrations are converted to $pCO_2$ (in µatm). Are the atmospheric values represent global mean or only southern hemisphere? Units have been changed to correspond directly with reference provided and "global" has been added to clarify region.

OC14: Line 221: Results and discussion on uncertainty of the seasonal mean are not clear (for me). Compared to measurements accuracy (2-5 µatm) these uncertainties seem small (5 µatm or less) suggesting that seasonal cycle is relatively well constrained with available observations. Correct? Measurement accuracy on underway $pCO_2$ systems is estimated to be ± 2µatm (Pierrot et al. 2009), and SOCAT QC flags included in this analysis list an accuracy of <5 µatm. The uncertainty of the seasonal mean presented in Figure 2 represent the standard error around the mean for the available years of data, by month. These errors are indeed small (around 5µatm). We have added further discussion into this section to clarify for readers and distinguish between measurement accuracy and the errors reported here.

Pierrot, D., Neill, C., Sullivan, K., Castle, R., Wanninkhof, R., Luger, H., Johannessen, T., Olsen, ¨ A., Feely, R. A., and Cosca, C. E.: Recommendations for autonomous underway pCO2 measuring systems and data reduction routines, Deep-Sea Res. Pt. II, 512–522, 2009.

OC15: Line 227: What is the origin of the $pCO_2$ peak in June (in both SPSS and PAZ)? An anomaly related to a specific location, cruises, year? Is such signal also seen in floats? Why this is not reproduced in SOM-FNN? The June peak in the SOCAT-noDP dataset seasonal cycle is

due to high pCO₂ observations reported in June 2008 just downstream of the Drake Passage (and therefore in the SOCAT-noDP dataset). The bar plots below the seasonal cycles in Figure 2 and Supplementary Figure 2 show the number of years of data available for each specific month. There are very few years available for June, and the data that is available for SPSS is occurs in years 2004 and 2008 only and is all located just downstream of the Drake Passage. Additionally, preliminary analysis of floats in the SPSS region do not show a consistent signal of a June peak however further analysis would be required to make broader conclusions for float seasonal cycles. The SOM-FFN product assembles data from around the globe and reconstructs them using monthly 1°x1° driver fields. In case of local events that are not represented in these "coarse" driver fields, the SOM-FFN method is likely to smooth out small scale and high frequency variability.

OC16: Line 230: Figure 3: I am surprised some data in SOCAT are not (apparently) seen in austral winter (JJA). Maybe change the color scale to better highlight the locations of the data (white-yellow not very clear on this figure). I would suspect authors used SOCAT-V4 for Cruises with Flag A, B, C, D, i.e. pCO₂ data from drifting buoys (flag E) not included. (however, later in this paper authors use pCO₂ derived from pH SOCCOM floats). If this is correct, authors should specify why they are not using all SOCAT-V4 data, including pCO₂ from 14 drifting buoys in the S.O. in 2002-2011 (and one launched in DP). We have changed the color bar on Figure 3 in order to improve the clarity of the figure. As stated above, and added to the manuscript, we now utilize SOCATv5 however we maintain only data with QC Flags A-D and do not include drifting buoys. One reason is to remain consistent with how SOCAT calculates their gridded products which many groups use for their analysis. Another reason is that data with QC flags A-D have uncertainty values <5 µatm as compared to estimated accuracy better than 10 µatm for flag E data. The portion of the paper that presents pCO₂ data calculated from SOCCOM floats is kept separate and used only as a comparison in this case and the higher uncertainty levels are clearly presented and discussed. Future work could continue this comparison between SOCCOM float observations and those of the drifting buoys in the Southern Ocean, specifically with regard to crossover instances between the two.

OC17: Line 255: For the anomaly in June could you explain why you "expect data to be less similar to that collected in the Drake Passage." This statement references results shown in Figure 4b, however give that this figure has not been presented or discussed yet at this point of the manuscript we have reworded the sentence to add clarification of our point.

OC18: Line 265: Authors indicate that "the Drake Passage seasonal cycle is representative of the broader SPSS biome seasonality, based on the available observations to date." However recent data from floats suggest high pCO₂ in winter, well above 400 µatm (Williams et al. 2017) and one would conclude that DP is not always representative of the SPSS. The statement referred to here is in reference to Figure 2, which shows that the seasonal cycles do not differ significantly for SOCAT data inside and outside of the Drake Passage region defined. We have added text to clarify that this does not take into account float data. Additionally, recent float data showing high pCO₂ in winter as mentioned, does not necessarily preclude that the DP is not representative of the SPSS as implied by the reviewer. In fact, the DPT frequently records observed pCO₂ higher than 400 µatm (Figure 7a,c).

OC19: Line 267: Again, in May-June, some data from buoys are available in SOCAT (values ranging between 320 and 400 µatm for these months). Are these data used in your study? As previously stated, only SOCAT data with flags A-D are included, thus excluding the buoy data referenced here. Additional analysis comparing buoy data to SOCCOM float data is an exciting next step of potential analysis but this analysis is focused on underway SOCAT/DPT data.

OC20: Line 270: Authors suggest that SOM-FNN is likely being driven by Drake Passage data. However in Figure 6 they show that SOM-FNN with DP or no-DP lead to the about same results. It would be interesting to show the seasonality derived from SOM-FNN with SOCAT-noDP (like for the trends presented in Figure 6). Figure 6 does indeed show that SOM-FFN does not report

significant differences with the inclusion/exclusion of Drake Passage data. The creation of the product allows for data from all regions of the global ocean to educate and help determine values for regions lacking data, given the training variables included. The statement that the reviewer is pointing to however is in regard to the SOCAT-DP, SOCAT-noDP, and SOCAT-all seasonal cycles. The statement merely states that because of a lack of data outside of the Drake Passage during certain months, the SOCAT-all seasonal cycle is likely being dominated by the data from the Drake Passage. The statement is not in reference to the SOM-FFN. We have edited these sentences to help clarify.

OC21: Line 273: To conclude on the seasonality, because authors construct a new "climatology" for SPSS based on data for the period 2002-2015, it would have been interesting to compare and discuss these cycles (DP and no-DP) with previous climatologies (e.g. Takahashi et al 2009, 2014) that used data back to the 80s (including winter cruises in the 90s in the SPSS but not used in the present study). Thank you for this suggestion. We are not constructing a "new climatology" as you say, but simply showing the average seasonal cycles for the different subsets of data in Figure 2. We do agree though that it would be interesting to compare to the Takahashi climatology that includes data dating back to the 1980s (and even before). The figure shown here is the same as Figure 2 but includes a bold black line representing the mean seasonal cycle for the SPSS biome from the Takahashi et al. 2009 climatology. The dashed black line is a corrected curve to year 2002 (assuming 1.95 µatm/yr trend as was used in this manuscript). While there are differences, the magnitude and general shape of the curve are similar to that of the SOCAT-all curve (blue).

[Figure]

OC22: Line 286: Could you really compare your results with the model of Lovenduski et al (2015)? The model was applied for a different period (1981-2007) and the SPSS biome in this model seems different (back to my comment on SPSS definition). We do not specifically "compare" these results to those of Lovenduski et al. 2015 but instead use the results from the model to motivate specific analysis on this data. Text has been added to highlight that the model is for a different subset of years. To highlight the low IAV in the model, would it be possible to plot the model results in DP like for observations (Figure 4a)? These results suggest that the model is not able to capture correct IAV and this is important to notice as such models are also used for prediction. Is your analysis in DP would help to identify processes that should be first revised in the model (in few words: dynamics, mixing, biology or others)? We thank the reviewer for this suggestion. We utilize the model here simply to motivate a discussion of interannual variability

and do not include any analysis of the model within this manuscript. The ideas presented are outside the scope of this paper and will be left to future model-based analyses to assess.

OC23: Line 292: To better follow the discussion on IAV, would be nice to add a figure, like Figure 4a but for SOM-FFN in DP. How your IAV results impact on the next steps, i.e. errors associated to long-term trends and fluxes analyses (Figure 6) and our understanding of the S.O. carbon sink variability? This figure would look nearly identical to current Figure 4a (see below). The increased smoothness is a factor of this being based on an interpolated product. We do not think that including this figure in the manuscript would add much to further the discussion within.

[Figure]

OC24: Line 325-330: All trends presented in Figure 5 are below atmospheric value. However, in previous analysis (and in the PAZ, Supp Figure 3) trends were near or above atmospheric level in north DP (Takahashi et al 2014; Munro et al 2015). These differences should be discussed (related to new data, region selected, years?). We have added further discussion on these differences. Results presented in Figure 5 should be listed in a table (or Supp. Mat.). We thank the reviewer for this comment and have added a table with the trend values. Another conclusion: the trends, about 1.5 µatm/yr, confirm the corrections applied in $pCO_2$ climatologies for reference year (Takahashi et al., 2009; 2014).

OC25: Line 350-355: Results presented in Figure 6 lead to several questions. First, as previous sections focused on SPSS biome, why Figure 6 now shows results for the Southern Ocean, south of 35°S? Are the results for SPSS only lead to the same conclusions? We thank the reviewer for this comment. We have changed Figure 6 to represent only the SPSS biome so as to remain consistent with the region of interest in the rest of the paper. Conclusions are not significantly changed. Second, it is interesting that results of the SOM-FNN with or without DP data lead to the same conclusion: the sink increases. Why the flux (and $pCO_2$ trends) are so close with or without DP data? We agree that this result is interesting and may surprise some. We believe this comes from the inherent properties of the product itself in that it utilizes data from around the globe to inform the values for regions without data in the Southern Ocean. We believe the SOM-FFN method has "learned" robust relationships between environmental driver data and $pCO_2$ and is therefore able to reproduce the $pCO_2$ variations as they are reflected in the environmental predictors (e.g. SST variations). Would that means that no-DP data are suitable to reproduce the increasing sink? Or is it because results are shown for all regions south of 35S (i.e. more data in the band 35-50S)? It would be interesting to show a map of the $pCO_2$ trends from both SOM-FNN with and without DP data to identify the main differences (if any). Trends are reported on Figure 6 for the SPSS biome (and previously for regions south of 35°S). On the other hand, the results

suggest stability in 2011-2015. I think this is new and should be discussed. What is the origin of the shift around 2011? This seems related to no-DP data (correct?). The exclusion/inclusion of Drake Passage data does not seem to be the cause of this shift beginning in year 2011 as both curves on the plot are consistent with regard to this shift. We agree that this is an interesting result and it deserves further investigation, however we feel that it is outside of the scope of this paper.

OC26: Line 370: Taking into account all uncertainties (standards, SST and equilibrium temperature calibrations,. . .) the accuracy of +/-2 µatm for ship-based systems is the best that can be achieved. Many cruises in SOCAT received a flag (C,D) with accuracy of +/-5 µatm (about half the floats, 11 µatm). Thank you for this comment. While it is true that SOCAT data with a flag C or D report accuracy better than 5 µatm, the comparison discussed in this section of the manuscript (Section 5) focuses *only* on Drake Passage Time Series data which has a reported uncertainty of +/-2 µatm.

OC27: Line 371: "there is great potential for these two observational platforms to work in concert". Yes, I fully agree. This could be done in this study, e.g. exploring the seasonality with merging product, or extend trend analysis to 2017. It is outside the scope of this analysis to combine SOCCOM float data into the dataset. As the reviewer mentioned earlier, it is a hotly discussed topic if SOCCOM (and other platforms) be included in SOCAT and specifically in the SOCAT gridded products. Work is being conducted on this front and we leave it to those groups to present their results in due time.

OC28: Line 374: If you use SOCAT-V5, other data are available in 2016. Might be useful to explore and add few more cross-overs with floats. The data used in the crossover plots is solely from the Drake Passage Time series and not from SOCAT (although they are not mutually exclusive). Broader comparisons of SOCAT data to SOCCOM floats is underway by other researchers whose manuscripts are in review (Gray et al. in review). Crossovers between the DPT and SOCCOM floats passing through the Drake Passage is the focus of this analysis.

OC29: Line 375-380: Figure 7 shows nicely when and where floats data are available, but comparison of underway $pCO_2$ and floats is not easy to see on figure 7. Authors should add a plot of $pCO_2$ versus time for the period 2016-2017 (e.g. like figure 5 in Williams et al., 2017). We thank the author for this suggestion. In a previous version we had a figure similar to what the reviewer suggested (See below for reviewer). The difficulty we faced is that we wanted to show the comparison of the DPT data to the floats with both the spatial and temporal spread as well as the $pCO_2$ comparisons. We feel that the current plots in Figure 7 accomplish this.

[Figure]

OC30: Line 380: pCO$_2$ data from floats in June-July 2017 suggest high pCO$_2$. What are the values (not clear in figure 7)? Are these values coherent with other floats in the SPSS in other sectors (float 9096, Williams et al., 2017). Floats in the Drake Passage region (those shown in Figure 7) with collected data in June and July 2017, report calculated pCO$_2$ ranging from 384.6µatm to 427.6µatm, with a mean of 401µatm. Underway Drake Passage Time Series pCO$_2$ measurements during that same timeframe (June/July 2017) collected measurements as high as 517µatm, but have an average of 373µatm. Figure 7 shows these comparisons of pCO$_2$ between floats and underway DPT.
Your question as to if the June/July 2017 high pCO$_2$ values from floats are coherent with other floats in the SPSS biome is not a topic that we will address in this manuscript as float intercomparison has been thoroughly analyzed by those from the SOCCOM group (Williams et al. 2017 and continued efforts). Also, to note- Float 9096 that you mention (not included in this manuscript) does not have reported pCO$_2$ values after February 2017.

OC31: Line 406-420: Authors recalled that pCO$_2$ derived from floats are subject to uncertainties associated to pH calibration, TA reconstruction, etc.. Here, they have in hand TCO$_2$ and pCO$_2$ data from DPT, and it would be interesting to quantify comparison for Salinity (used for TA), TA and pH (from DPT TCO$_2$/pCO$_2$ data). Would this lead to the same accuracy (+/- 11 µatm). What TA algorithm is used in your calculations? Would it be appropriate to use an algorithm specific of the DP region? We utilize the LIAR algorithm for TA calculations from SOCCOM float data (Carter et al. 2016). In response to the reviewers other ideas mentioned here, indeed they are great questions worthy of investigation in a follow-on study. Below are a few preliminary plots showing comparisons of underway DPT data and SOCCOM float Total Alkalinity, SST, and pH for the defined Drake Passage area.

Carter, B. R., N. L. Williams, A. R. Gray, and R. A. Feely (2016), Locally interpolated alkalinity regression for global alkalinity estimation, *Limnol. Oceanogr. Methods*, *14*(4), 268–277, doi:10.1002/lom3.10087.

[Figure]

[Figure]

[Figure]

**Drake Passage Total Alkalinity - DPT and SOCCOM floats**

OC32: Line 415: Authors recalled that in previous analysis, mean difference with ship data was only 3.7 µatm. What mean difference did you get in your comparison at DP? The mean difference referenced by the reviewer is a comparison between the first float profile and underway data collected "nearby". The analysis shown in Figure 8b is similar but is not restricted to being on the first float profile. As Figure 8b shows, there are multiple underway observations within the designated crossover criteria. Mean differences for each float range from -11.95 µatm to 11.77 µatm for the crossovers shown in Figure 8b.

OC33: Line 426-427: Authors conclude: "With this complete coverage we find seasonal amplitudes in the SPSS to be smaller than subpolar regions in the Northern Hemisphere, and controlled by a combination of temperature and deep water mixing effects". I think this is not really new but confirm previous results (e.g. Takahashi et al 2009; Rödenbeck et al., 2015) and authors did not evaluate processes (temperature, mixing) in their study. Thank you for this comment. We have revised this paragraph with the reviewer's comments in mind.

OC34: Line 426-427: Authors conclude: "Uncertainties in the seasonality remain considerable". What is "considerable"? Would the conclusion the same if all SOCAT data are used, including data back in the 90s with more cruises conducted in winter? We have revised this paragraph with the reviewer's comments in mind. We do not feel the uncertainties in seasonality would be much improved with data from the 1990s as such limited full-seasonal coverage data is available in the subpolar Southern Ocean prior to the beginning of the DPT project.

OC35: Line 430: do we need such analysis to conclude that there is a lack of winter data? This is an important message for future observations, but for the past decades the only way to mimic winter data is to reconstruct at best $pCO_2$ fields (such as SOM- FNN, or ocean models). For the Southern Ocean, I understand that the $CO_2$ sink and its decadal variability is relatively well-known (e.g. Landschützer et al., 2015 and your figure 6) although winter data were sparse. We stand by this statement and the necessity for additional winter data. Additional winter-time data outside of the Drake Passage is essential in order to identify if the seasonality of the waters within the Drake Passage and in the broader subpolar region are consistent, specifically the winter-time peak that

is shown in the SOCAT-noDP analysis.

OC36: Line 445-447: Authors conclude: "Southern Ocean has been a growing sink for atmospheric carbon since 2002." Again, from your results (Figure 6) it appears that ocean $pCO_2$ increased faster in recent years (2011-2015) compared to 2002-2011, and fluxes relatively stable. This result, not discussed in the manuscript, argues to maintain long-time series observations to better detect how (and why) the ocean $pCO_2$ is changing, and subsequently the fluxes in this important region. A thorough investigation of this transition is beyond the scope of this paper. Considering the full period (2002-2016) the Southern Ocean has been a growing sink, hence we have not made a change to this statement.

OC37: Line 449: Authors conclude: "Comparisons between underway DPT and SOCCOM float measurements show general agreement". What is "general agreement" ? Please give a number (e.g. mean differences). Are your comparisons, here specifically for the DP region, confirm or not previous analyses (difference around 4 µatm, Williams et al., 2017)? We have reworded this sentence to be more specific. Please refer to the response to OC32 which describes mean differences between the floats and DPT observations as compared to that found in Williams et al. 2017. Figure 8 clearly shows the differences for when crossovers occur between these two platforms.

OC38: Line 452: Authors conclude: '...could aid in reducing the uncertainty on the float $pCO_2$ measurements by helping to identify problematic float sensors.". In your analysis, did you experienced problem with sensors? Our research does not explore sensor issues. The high uncertainty levels on float $pCO_2$ is due to compounding uncertainty from multiple calculations that go into acquiring the necessary parameters to calculate $pCO_2$. Additionally, instrument drift and failure is an issue that could be improved upon through technological developments. These improvements could help to reduce the uncertainty in float calculated $pCO_2$ which would improve comparison efforts such as this one by better and more directly being able to compare shipboard observations and float estimates.

OC39: Line 454 : Authors conclude that a coordinated monitoring efforts that combine underway and float data is highly needed. I totally agree. If achieved, this would be a very important step, not only for the Southern Ocean as a test experience, but should be engaged at global scale (a new dream). In this context, and authors have the data in hand, I think the present study should present results combining underway and floats data (at least for the seasonality in DP, maybe for the trends 2002-2017 as a first test). We thank the reviewer for this comment and agree that such a coordinated monitoring effort would be ideal. We feel that this study is a first step in utilizing crossover comparisons and acknowledge that other groups with more direct knowledge of the float sensors are also investigating these comparisons. This research definitely could pave the way for future comparisons. We have added a discussion of seasonality comparisons between the Drake Passage Time series data and SOCCOM float estimated $pCO_2$ but a more in-depth comparison is out of the scope for this manuscript. A thorough investigation of these comparisons with multiple shipboard time series globally would constitute enough discussion for another manuscript.

OC40: Line 535: Correct reference: Gruber et al 2009 We have corrected the reference.

OC41: Line 700: Correct reference: Williams et al 2017 We have corrected the reference.

OC42: Figure 1: add name of other colored biomes (ICE)? This map might be extended to 35S (the limit chosen for figure 6)? We have updated Figure 1 with new colors and labels.

OC43: Figure 3: Legend: add that gray lines depict the SPSS biome. Change color- scale (white-yellow not very clear). We have updated the colorbar for this figure and added text in the caption to identify the gray lines.

OC44: Figure 7: legend 7c: the plot starts from Jan-2016 (not Oct 2015) We have now corrected the start year accordingly in the figure legend

---

## Author Comment (AC2) · 11 May 2018

We thank the reviewer for their comments and suggestions. As for SOCCOM data quality specifics, we direct the readers to Johnson et al. 2016, Johnson & Claustre 2016, and Williams et al. 2017 specifically for pCO2, as these publications are very thorough regarding sensors and the quality of each variable measured by the floats.

In a previous version of this manuscript we had a figure showing a simple time series presenting the overlap period of underway DPT data and SOCCOM float calculated pCO2 (See attached, for reviewer). The difficulty we faced is that we wanted to show the comparison of the DPT data to the floats with both the spatial and temporal spread

[Figure]

as well as the pCO2 comparisons. We feel that the current plots in Figure 7 accomplish this. We do now expand on our comparison of seasonality specifically and we have added further discussion on this topic in Section 5.

[Figure]

[Figure]

[Figure]

**Fig. 1.**

---

## Author Response (AR1)

Dear Editor,

Thank you for your recommendation for us to revise our manuscript for resubmission. We have responded to each of the 2 reviewers comments and questions individually in the responses posted online. Overall we have updated the manuscript with the most up-to-date version available for each dataset. Additionally, specific changes include an additional figure (Figure 7) that investigates seasonality of the floats and DPT values as well as further discussion on these comparisons in Section 5 of the manuscript. Throughout the manuscript we have elaborated on comparisons to references, specifically for our discussion of trends.

We look forward to hearing from you regarding the next steps for publication.

On behalf of my coauthors
Amanda Fay

[revised manuscript text omitted]

Amanda Fay 5/27/2018 1:28 AM

Amanda Fay 5/27/2018 1:28 AM

Amanda Fay 5/27/2018 1:28 AM

Amanda Fay 5/27/2018 1:28 AM

Amanda Fay 5/27/2018 1:28 AM

Amanda Fay 5/27/2018 1:28 AM

Amanda Fay 5/27/2018 1:28 AM

Amanda Fay 5/27/2018 1:28 AM

Amanda Fay 5/27/2018 1:28 AM

variability using data from the DPT, it is important to document how pCO$_2$ in this particular region compares with pCO$_2$ measured elsewhere in the subpolar Southern Ocean. In this study, we utilize available ship-based surface ocean pCO$_2$ observations collected in the subpolar Southern Ocean to evaluate

130  how the seasonal cycle, interannual variability, and long-term trends of surface ocean pCO$_2$ in the Drake Passage region compare to that of the broader subpolar Southern Ocean. Further, we highlight the opportunity for post deployment assessment of autonomous observational platforms passing through the Drake Passage utilizing the high frequency, underway pCO$_2$ measurements from the DPT.

135  **2. Data**

This study uses several observational datasets and data products of surface ocean pCO$_2$ in the Southern Ocean: measurements from the Surface Ocean CO$_2$ Atlas (SOCAT), which includes underway measurements from the DPT, interpolated estimates of the SOCAT data using a self-organizing map feed-forward neural network (SOM-FFN) approach, and calculated pCO$_2$ estimates from biogeochemical Argo

140  floats. While the SOCAT database reports the fugacity of carbon dioxide (fCO$_2$), for our analysis we consider datasets reporting pCO$_2$ and fCO$_2$ to be interchangeable. This is an acceptable assumption for surface ocean observations as CO$_2$ behaves closely to an ideal gas. Globally, the difference between these parameters is less than 2 μatm, with fCO$_2$ being smaller than pCO$_2$ by no more than 2 μatm due to temperature dependence. This is roughly the reported uncertainty of shipboard observations of pCO$_2$ and

145  well within the uncertainty of the observation-based pCO$_2$ estimates. Below, we describe each of these data sources in turn.

**2.1 The Drake Passage Time-series (DPT)**

A unique dataset of ongoing year-round observations beginning in 2002 is available from the Drake

150  Passage Time-series. This data set provides an unprecedented opportunity to characterize the mean and time-varying state of the Drake Passage and surrounding waters using direct observations. In addition to high frequency underway observations of surface ocean pCO$_2$, other physical and biogeochemical variables measured onboard allow for a complete understanding of the carbonate system in the Drake Passage.

Amanda Fay 5/27/2018 1:28 AM

Amanda Fay 5/27/2018 1:28 AM

Amanda Fay 5/27/2018 1:28 AM

Amanda Fay 5/27/2018 1:28 AM

Amanda Fay 5/27/2018 1:28 AM

Amanda Fay 5/27/2018 1:28 AM

Amanda Fay 5/27/2018 1:28 AM

Amanda Fay 5/27/2018 1:28 AM

Amanda Fay 5/27/2018 1:28 AM

Amanda Fay 5/27/2018 1:28 AM

Amanda Fay 5/27/2018 1:28 AM

Amanda Fay 5/27/2018 1:28 AM

[revised manuscript text omitted]